# Rockfall trajectory reconstruction: A flexible method utilizing video footage and high-resolution terrain models

François Noël[1,2], Michel Jaboyedoff[1], Andrin Caviezel[3], Clément Hibert[4], Franck Bourrier[5], Jean-Philippe Malet[4,6]

[1]Risk Analysis Group, Institute of Earth Sciences, University of Lausanne, CH-1015 Lausanne, Switzerland
[2]Geohazard and Earth Observation, Geological Survey of Norway – NGU, NO-7040 Trondheim, Norway
[3]WSL Institute for Snow and Avalanche Research SLF, CH-7260 Davos, Switzerland
[4]ITES/Institut Terre et Environnement de Strasbourg, CNRS UMR7063 CNRS – Université de Strasbourg, 5 rue Descartes, F-67084 Strasbourg, France
[5]Université Grenoble Alpes, INRAE, ETNA, 38000 Grenoble, France
[6]École et Observatoire des Sciences de la Terre, CNRS UAR 830 CNRS – Université de Strasbourg, 5 rue Descartes, F-67084 Strasbourg, France

*Correspondence to*: François Noël (francois.noel@ngu.no)

**Abstract.** Many rockfall simulation software provide great flexibility to the user at the expense of a hardly achievable parameter unification. With sensitive site-dependent parameters that are hardly generalizable from the literature and case studies, the user must properly calibrate simulations for the desired site by performing back calculation analyses. Thus, rockfall trajectory reconstruction methods are needed. For that purpose, a computer-assisted videogrammetric 3D trajectory reconstruction method (CAVR) built on earlier approaches is proposed. Rockfall impacts are visually identified and timed from video footage and are manually transposed on detailed high-resolution 3D terrain models that act as the spatial reference. This shift of reference removes the dependency on steady and precisely positioned cameras, ensuring that the CAVR method can be used for reconstructing trajectories from witnessed previous records with nonoptimal video footage. For validation, the method is applied to reconstruct some trajectories from a rockfall experiment performed by the WSL Institute for Snow and Avalanche Research SLF. The results are compared to previous ones from the SLF and share many similarities. Indeed, the translational energies, bounce heights, rotational energies and impact positions against a flexible barrier compare well with those from the SLF. The comparison shows that the presented cost-effective and flexible CAVR method can reproduce proper 3D rockfall trajectories from experiments or real rockfall events.

## 1 Introduction

Many rockfall simulation software provide great flexibility to the user at the expense of a hardly achievable parameter unification, as highlighted by Berger and Dorren (2006), Berger et al. (2011), Volkwein et al. (2011), Jarsve (2018), Garcia (2019), Bourrier et al. (2021), and Noël et al. (2021). Even when using the same rebound model, though it may be implemented in a different software, the results using the same parameters may vary, as shown previously in Noël et al. (2021) when comparing CRSP 4 (Pfeiffer and Bowen, 1989; Jones et al., 2000) with RocFall 8 (Stevens, 1998; Rocscience

Inc., 2022). The settings of the rebound model parameters are often specific to each model, rockfall software and version used. Thus, it is difficult to transpose them from experimental results, such as the apparent coefficient of restitution from impact experiments. Indeed, even if rebound model parameters are classically called "coefficients of restitution" (e.g., $R_N$ and $R_T$ for the normal and tangential components), they are not the same as the apparent coefficients of restitution (e.g., $COR_N$ and $COR_T$ for the respective components) and cannot be directly interchanged, as explained in Noël et al. (2021). For example, a rebound calculated with the model of Pfeiffer and Bowen (1989) using a normal "coefficient of restitution" of $R_N$ = 0.35 as damping parameter returns a low normal translational velocity of 1.6 m s$^{-1}$ for a normal incident velocity of 10 m s$^{-1}$. In that case, the ratio of returned velocity over the incident one gives a calculated apparent coefficient of restitution of $COR_N$ = 0.16, which is different than the $R_N$ of 0.35 used. Because it is difficult to transpose the simulation parameters from experimental results, finding the proper parameter values are thus far more limited to a range of rock sizes, shapes, terrain materials and saturation, perceived roughness, and profile geometry. They are then hardly generalizable from the literature or from one simulation model to another and are often site dependent, as highlighted by Volkwein et al. (2011) and Valagussa et al. (2015) and shown by the variable benchmark results in Berger and Dorren (2006), Berger et al. (2011), Garcia (2019) and C2ROP (2020).

Therefore, it is often emphasized to properly calibrate simulations for the desired site by performing back calculation analyses on similar sites and from on-site rockfall experiments (Jones et al., 2000; Labiouse, 2004; Berger and Dorren, 2006; Berger et al., 2011; Volkwein et al., 2011; Valagussa et al., 2015; Bourrier et al., 2021; Noël et al., 2021). For that purpose, it is necessary to evaluate the main trajectory paths of the rockfalls, their runouts, how the velocities, bouncing heights and energies evolve along these paths, as well as after each impact and how the rocks deviate laterally. This raises the need for cost-effective and flexible 3D trajectory reconstruction methods to help gather and share the data needed for a site-specific calibration of rockfall simulations. Additionally, the gathered data could later be used for the improvement and development of more objective rockfall simulation methods that are less dependent on the inconvenient and expensive need to perform back analyses.

As illustrated by Volkwein and Klette (2014) and Caviezel et al. (2019), different methods exist for reconstructing rockfall trajectories. Some reconstructed parts of the trajectories in 2D as seen from above (e.g., Volkwein and Klette, 2014; Volkwein et al., 2018), others in 2D vertical profiles (e.g., Glover et al., 2012; Wyllie, 2014; Spadari et al., 2012; Bourrier et al., 2012). Few reconstructed the trajectories in the 3D space and documented their lateral deviations (e.g., Dorren et al., 2005; Dorren and Berger, 2006; Dewez et al., 2010; Hibert et al., 2017; Caviezel et al., 2019; Bourrier et al., 2021). From them, Dorren et al. (2005) and Dorren and Berger (2006) used rangefinders with a tiltmeter and a compass to measure the position of each impact, requiring time-consuming and potentially exposed field work to obtain the valuable field data. Dewez et al. (2010) also reconstructed trajectories in 3D, but this time, the rock positions were remotely estimated from video footage using the cameras as references. For that, their method required precisely synchronized and undistorted video pairs captured with a wide field of view (FOV) of ~76° from precisely positioned steady cameras. With the help of a script, the center mass of the falling rocks was manually located frame-by-frame on the displayed video pairs. The use of a

relatively high frame rate (50 fps) gave a good time resolution of 1/50$^{th}$ of a second for precision, but it increased the number of frames on which to perform the manual tracking of the rocks.

This time-consuming manual process can be partly automated based on the method proposed by Caviezel et al. (2019),
increasing the objectivity of the reconstruction process. This is done by producing dense 3D point clouds by photogrammetry from each synchronized undistorted frame of steady video footage captured from different viewpoints. The 3D points corresponding to the visible side of the artificial rocks facing the cameras are then extracted based on their contrasting artificial painted colors compared to the background. The center of mass of the rocks is estimated from the convex hulls formed by meshing the extracted 3D points.

Compared to Dewez et al. (2010), this automation process can introduce an erroneous shift of the reconstructed center of mass toward the cameras if the 3D points of the occluded backside of the rocks not visible by the cameras are missing. However, this can be workaround by fitting 3D models of the controlled rock shapes on their partial photogrammetric reconstruction. Additionally, ultrahigh resolution (e.g., 8K UHD in Caviezel et al., 2019) and sharp contrast of the falling rocks with their backgrounds are needed for feature recognition to compensate for the relatively wide FOV needed for
framing the whole site from each fixed viewpoint. Due to recording data rate constraints, ultrahigh resolutions and raw footage can limit the recording frame rate depending on the acquisition equipment (e.g., 25 fps in Caviezel et al., 2019), thus reducing the time resolution and related precision. Consequently, the method requires relatively high-end camera bodies coupled with proper sharp lenses and powerful computers for processing the associated data, producing thousands of frame-by-frame dense 3D point clouds and aligning them.

Despite being partly automated, the time-consuming processing complexifies the iterative visual validation that the reconstructed trajectories match with reality and fine-tuning processes following the first reconstructions. As a result, numerous reconstructed impacts with an energy balance above 1.00 involving apparent gain of kinetic energy can be obtained with this method, sometimes with an increase for both the translational and angular velocities after impact, as shown in Caviezel et al. (2019, 2021). These abnormal impacts can be explained by energy transfers from the height
differences between the beginning and the end of the impacts with long rock-ground interactions (Caviezel et al., 2019, 2021). As later shown, this may also be attributed to timing and positioning imprecisions, especially for impacts with short rock-ground interactions.

In this work, an alternative cost-effective and flexible computer-assisted videogrammetric 3D trajectory reconstruction method (CAVR) is proposed. It can be used in addition to the aforementioned approaches, as it brings complementary
information when the video footage is not optimal, when automatic tracking is not possible or if abnormal apparent kinetic energy gain is observed at impact. The improved method is built from the concepts of the previous methods, and it was preliminarily tested in Noël et al. (2017, 2018). It involves a computer-assisted manual tracking of rocks and a high frame rate (e.g., 120 fps) for a precise time resolution as in Dewez et al. (2010). The time-consuming tracking of the freefalling phases is, however, avoided, as this phase can be accurately and efficiently reconstructed from ballistic equations, as in

Volkwein et al., (2011), Wyllie (2014), Glover (2015), Gerber (2019), and similar to the method used in Bourrier et al. (2012) and Hibert et al. (2017).

As in Caviezel et al. (2019), the proposed CAVR method relies on the use of 3D models to estimate the position of the rocks. However, instead of generating thousands of frame-by-frame 3D photogrammetric models of the rocks, the proposed method uses one detailed textured 3D terrain model for the spatial reference coupled to the efficient 3D point cloud impact detection

algorithm by Noël et al. (2021). The algorithm is used to locate with proper offset the center of mass of the rocks above the ground at impacts. Contrary to the tracking methods of Dewez et al. (2010) and Caviezel et al. (2019), the cameras can be zoomed to narrow FOVs and move or panned to track the rocks, since the 3D detailed terrain model acts as the spatial reference instead of the cameras. This produces detailed close-up footage of the rocks and the surrounding terrain features that facilitate the visual identification of the impact points with the ground. Lens distortion is less problematic as it shifts

equally the captured rocks with their surrounding terrain that acts as reference. It also increases the flexibility of the method, as different video footage can be used as input. Additionally, it reduces its cost by avoiding the need for high-end cameras and related processing equipment. The computer-assisted reconstruction process is semiautomatic, and the user obtains a real-time update of the 3D reconstructed freefalling parabolas forming the trajectory at the center of mass of the rock projectile properly offset from the ground. The impact point on the ground can be updated in real-time following the mouse

cursor on the screen. This incorporates the important visual validation of the reconstructed trajectories and iterative fine-tuning processes directly as part of the reconstruction process. This ensures reconstructing dissipative impacts (without apparent gain of kinetic energy) for impacts with short rock-ground interactions, as later detailed.

The proposed CAVR method relies on two inputs: the impact positions and their related time. The impacts are visually identified and timed from the high frame rate video footage, and they are manually transposed on a detailed corresponding

3D terrain model to obtain the 3D coordinates of their positions. In this paper, the common ballistic equations used for reconstructing the trajectories from these two inputs are first given with the other equations related to the different reconstructed values. Then, since the video footage is a central piece for the method, especially if there is no impact mark on the ground to act as a guide, the details about the acquisition of the video footage and the related precision and accuracy are meticulously described. This is followed by short subsections concerning the 3D terrain model, the rock block geometric

characteristics and the validation of the reconstructed trajectories. A developed computer tool incorporating the described concepts to assist and homogenize the reconstruction process is then described. A comparison of methods is finally presented and discussed.


**Table 1 – List of variables for the rockfall ballistics.**

| Variables | Description | Units | Variables | Description | Units |
|---|---|---|---|---|---|
| $t$ | Time | s | $E_k$ | Total kinetic energy of the rock | J |
| $\vec{X}$ | Position of the rock in 3D space | m | $\vec{N}$ | Vector normal to the ground surface orientation | m |
| $\vec{v}$ | Translational velocity of the rock | m s$^{-1}$ | $\theta_1$ | Incident impact angle with the ground | ° |
| $\overrightarrow{v_T}$ | Tangential component of the velocity | m s$^{-1}$ | $\theta_2$ | Returned impact angle with the ground | ° |
| $\overrightarrow{v_N}$ | Normal component of the velocity | m s$^{-1}$ | $\Delta\theta_{trend}$ | Delta of the trend direction of the incident trajectory and the aspect/dip direction of the terrain | ° |
| $\vec{\omega}$ | Angular velocity | rad s$^{-1}$ | $\theta_N$ | Deviation of the incident and returned $\overrightarrow{v_T}$ (around $\vec{N}$ axis) | ° |
| $\vec{g}$ | Acceleration of the rock | m s$^{-2}$ | $\theta_{dev}$ | Total deviation that the rock undergoes by the impact | ° |
| $m$ | Mass of the rock | kg | $COR_v$ | Total kinematic coefficient of restitution | - |
| $I$ | Moment of inertia of the rock | kg m² | $COR_T$ | Tangential kinematic coefficient of restitution | - |
| $p$ | Translational momentum of the rock | kg m s$^{-1}$ | $COR_N$ | Normal kinematic coefficient of restitution | - |
| $L$ | Angular momentum of the rock | kg m² s$^{-1}$ | | | |

## 2 Rockfall ballistics

Rockfall trajectories can be reconstructed from the impact positions and the associated time. This section details the ballistic equations required for the reconstruction of the 3D trajectories, related angles, velocities, kinetic apparent coefficient of restitution, momentum, and energies.

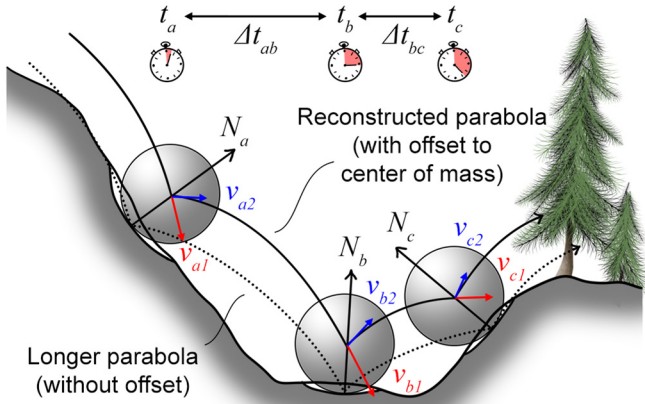

**Figure 1: Impact configurations for the reconstructed parabolas. Note how the offset of the impacts minimizes the common issues associated with the exaggerated parabola's lengths of impacts simplified to single points.**

The airborne 3D rockfall trajectory segments are a consecution of oblique throws, and their parabolic nature has been described by Galileo Galilei (Drake and MacLachlan, 1975). The position, translational velocity, and acceleration of a rock during its ballistic (freefalling) phase is defined by Eq. (1), (2) and (3) as follows:

$$\overrightarrow{X_t} = \overrightarrow{X_{t0}} + \overrightarrow{v_{t0}}t + \frac{1}{2}\overrightarrow{g_t}t^2 , \tag{1}$$

$$\overrightarrow{v_t} = \frac{d\overrightarrow{X_t}}{dt} = \overrightarrow{v_{t0}} + \overrightarrow{g_t}t , \tag{2}$$

$$\overrightarrow{g_t} = \frac{d\overrightarrow{v_t}}{dt} , \tag{3}$$

When neglecting drag due to air resistance (see Appendix A about the significance of air drag), the acceleration components are written as follows (Eq. 4):

$$\overrightarrow{g_t} = \begin{bmatrix} g_{xt} \\ g_{yt} \\ g_{zt} \end{bmatrix} \cong \begin{bmatrix} 0 \\ 0 \\ -9.81 \end{bmatrix} , \tag{4}$$

Then, the incident and the returned translational velocity are estimated for each impact from the previous equations. For that, the position of the rock's center of mass must be known for a series of successive impacts. The rock-ground interaction periods must also be very short relative to the freefall periods to ensure that they can be considered as impulses (Wyllie, 2014). Note that impacts simplified to single points require short rock-ground interaction periods. Most problems related to

the single point methods due to incorrect path lengths are minimized by offsetting the points to the center mass of the rock projectiles (Fig. 1). The velocities of an impact $b$ are preceded by an impact $a$ and followed by an impact $c$ as in Fig. 1, which is given by Eq. (5) and (6) as follows:

$$\overrightarrow{v_{b1}} = \begin{bmatrix} v_{xb1} \\ v_{yb1} \\ v_{zb1} \end{bmatrix} \cong \begin{bmatrix} v_{xa2} \\ v_{ya2} \\ v_{za2} - 9.81\Delta t_{ab} \end{bmatrix} ,$$

$$\tag{5}$$

$$\overrightarrow{v_{b2}} = \begin{bmatrix} v_{xb2} \\ v_{yb2} \\ v_{zb2} \end{bmatrix} \cong \begin{bmatrix} \Delta X_{xbc}/\Delta t_{bc} \\ \Delta X_{ybc}/\Delta t_{bc} \\ \frac{\Delta X_{zbc}}{\Delta t_{bc}} + \frac{1}{2}9.81\Delta t_{bc} \end{bmatrix} ,$$

$$\tag{6}$$

The translational and angular momenta $p$ and $L$ are given by Eq. (7) and (8) as follows:

$$p = mv , \tag{7}$$

$$L = I\omega , \tag{8}$$

The total kinetic energy is given by Eq. (9) as follows:

$$E_k = \frac{1}{2}mv^2 + \frac{1}{2}I\omega^2 , \tag{9}$$

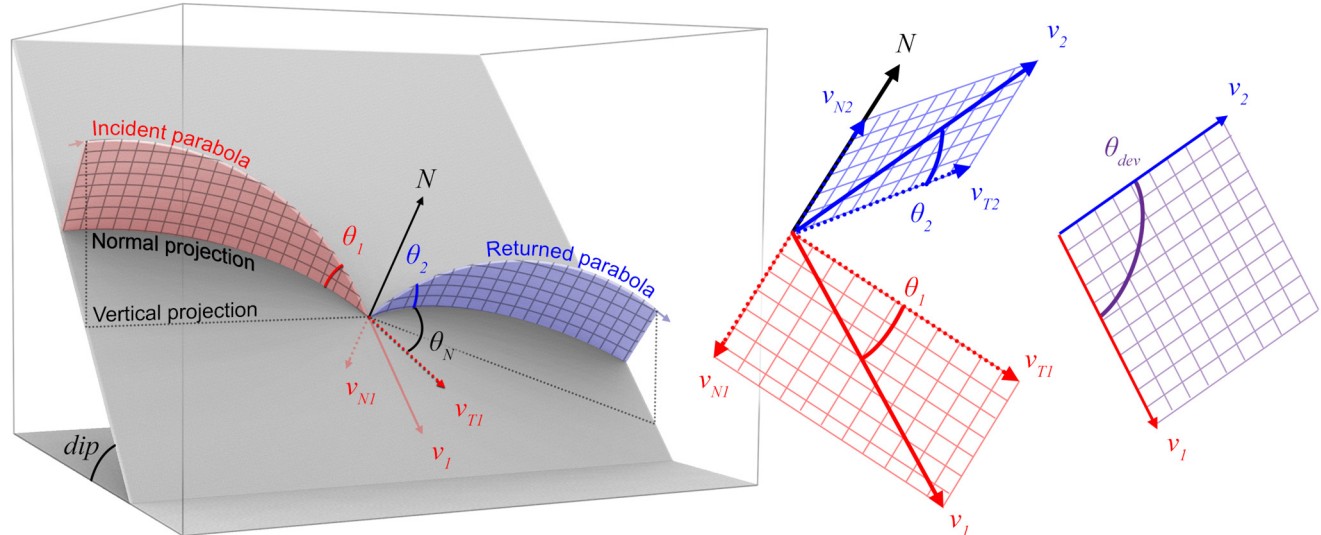

**Figure 2: Geometric configuration at impact of the reconstructed translational velocity vectors and the related angles (see Table 1 for the variable's descriptions). Note that such angles are measured based on the normal vector to the terrain (N) and not to the vertical. For a perfectly reflected impact, such as a light beam on a flat mirror, the incident and the returned velocity vectors ($v_1$ and $v_2$) would be coplanar with N. Here, the lateral deviation ($\theta_N$) is measured as the angle around N making $v_2$ deviate from the previous coplanar situation. Note that this rotation axis should be tilted slightly toward where the rock projectile is from depending on the amount of scarring, slipping, and skidding, but it is kept around N for simplicity. The total deviation ($\theta_{dev}$) is simply measured as the angle between $v_1$ and $v_2$. It is close to the sum of the incident and the returned angles ($\theta_1$ and $\theta_2$) when $\theta_N$ is small.**

Apparent coefficients of restitution can be calculated for each impact from the components of the obtained velocities (Fig. 2). They also correspond to the ratio of momentum preserved by the rock projectile after each impact. One should not use them directly as parameters for rockfall simulations since they generally do not correspond to the parameters used in the rebound models as mentioned in introduction and described in Noël et al. (2021). The total, tangential and normal apparent kinematic coefficients of restitution are given by Eq. (10), (11) and (12) as follows:

$$COR_v = \frac{\|\vec{v_2}\|}{\|\vec{v_1}\|}, \tag{10}$$

$$COR_T = \frac{\|\vec{v_{T2}}\|}{\|\vec{v_{T1}}\|}, \tag{11}$$

$$COR_N = \frac{\|\vec{v_{N2}}\|}{\|\vec{v_{N1}}\|}, \tag{12}$$

The rock-ground geometric configuration at impact can be analyzed simply with vector dot products. The incident and the returned impact angles with the ground (Fig. 2) are given by Eq. (13) and (14) as follows:

$$\theta_1 = sin^{-1}\left(\frac{|\vec{v_1} \cdot \vec{N}|}{\|\vec{v_1}\|\|\vec{N}\|}\right), \tag{13}$$

$$\theta_2 = sin^{-1}\left(\frac{|\vec{v_2} \cdot \vec{N}|}{\|\vec{v_2}\|\|\vec{N}\|}\right), \tag{14}$$

The angular difference in the horizontal plane between the trend direction of the incident velocity projected on the plane $(\overrightarrow{v_{xy1}})$ and the aspect direction of the terrain face from the normal projected on the plane $(\overrightarrow{N_{xy}})$ is given by Eq. (15) as follows:

$$\Delta\theta_{trend} = cos^{-1}\left(\frac{|\overrightarrow{v_{xy1}} \cdot \overrightarrow{N_{xy}}|}{\|\overrightarrow{v_{xy1}}\|\|\overrightarrow{N_{xy}}\|}\right), \tag{15}$$

The rock lateral deviation from a "perfectly reflected" rebound, i.e., the lateral deviation making $\overrightarrow{v_2}$ deviate from being coplanar with $\overrightarrow{v_1}$ and $\overrightarrow{N}$, is measured by a rotation around the normal vector axis, and is given by Eq. (16) as follows:

$$\theta_N = \pm cos^{-1}\left(\frac{|\overrightarrow{v_{T1}} \cdot \overrightarrow{v_{T2}}|}{\|\overrightarrow{v_{T1}}\|\|\overrightarrow{v_{T2}}\|}\right), \tag{16}$$

$\theta_N$ is set to negative if this deviation brings the azimuth of $\overrightarrow{v_2}$ closer to that of $\overrightarrow{N}$ or to positive if this deviation brings the azimuth of $\overrightarrow{v_2}$ away, as shown in Fig. 2.

The rock's total deviation due to the impact is given by Eq. (17) as follows:

$$\theta_{dev} = cos^{-1}\left(\frac{|\overrightarrow{v_1} \cdot \overrightarrow{v_2}|}{\|\overrightarrow{v_1}\|\|\overrightarrow{v_2}\|}\right), \tag{17}$$

## 3 Trajectory reconstruction method

As defined with the previous equations, it is possible to reconstruct rockfall trajectories with their velocity in between recorded impacts with a short rock-ground interaction period. The impact time and position are visually evaluated from video footage of rockfall events. The impact positions are then transposed on a 3D detailed high-resolution terrain model used as the spatial reference to retrieve their precise coordinates. Such coordinates on the ground need to be offset to the center of mass of the rocks, requiring the acquisition of the rock block geometry. Finally, the reconstructed trajectories must be visually validated, ensuring that they are aligned with the falling rocks. Such steps and inputs are detailed in this section.

### 3.1 Video footage

Because the detailed 3D terrain model is the spatial reference for the position of the impact, the reconstruction method is not constrained to the use of a special type of video footage or steady cameras with fixed viewpoints or lenses without distortion. Thus, it is well suited for reconstructing trajectories from previously witnessed records to help gather the data needed for site-specific calibration of sensitive rockfall simulations. Good video footage for this method is any footage where the position of the impacts can be visually located and timed. Therefore, "zoomed" footage with a narrow field of view (FOV), a manual panning to track the falling rocks, a high captured frame rate, and a high resolution adapted to the "sharpness" given by the acutance and resolving power of the lens is ideal for obtaining the most precision out of the method. The precision and the related acquisition setup concepts are detailed in the following subsections.

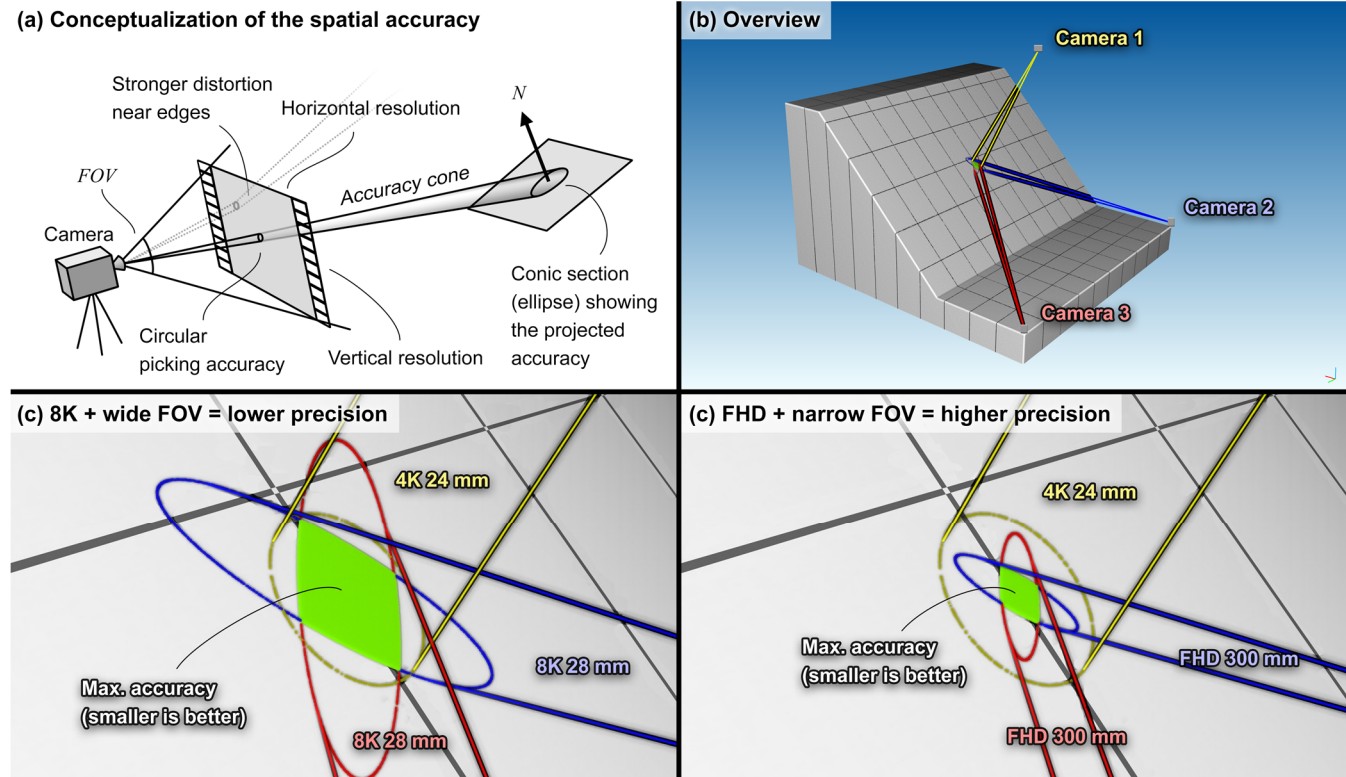

**Figure 3:** Conceptualization of the spatial accuracy from projecting the picking accuracy to the terrain. The picking accuracy is degraded to ±100 px to better illustrate the concept. In reality, the precision and accuracy depend on the resolving power and the acutance of the lens used, sensor size and resolution, general quality and sharpness of the footage and ease at distinguishing the rock projectile from the background. The elongation of the resulting conic section ellipses depends on the viewing "incident angle" with the terrain. The accuracy is maximal when the ellipses are small and not elongated, but they can also be maximized by combining two or more viewpoints. The size of the projected pixels depends on the resolution and the FOV used. The local accuracy can be better, even at a lower resolution, by using lenses with a small FOV. This also helps to distinguish the rock from the background, improving the picking precision and accuracy, which in turn also improves the local precision and accuracy.

### 3.1.1 Precision and accuracy

The optimal video footage for this method is any footage where a series of successive impacts can be visually located and timed. The sharper and detailed the image around the impact point is, the easier it is to precisely visually locate and time it. The resolving power of a camera system or of a lens attached to the camera body can be measured with the modulation transfer function (MTF). This optical performance measurement is often expressed as the number of alternating black and white line pairs (lp) that can be resolved on one millimeter of a camera sensor or film at a given contrast (Rowlands, 2020). The more lines that are captured, the finer the details that can be captured are. The more detailed the images captured, the more accurate and precise it is for transposing the impact positions when picking their position on the detailed terrain model. The circular area of the picking accuracy ($p_r$ [px]) around the impact point is analogous to the error bars around a point; the shorter the error bars (i.e., the radius of the circular area) are, the better (Fig. 3). The corresponding spatial accuracy on the terrain can be obtained by projecting this circular area to the terrain, following the reversed paths of the light rays that

reached the camera sensor during the video capture (Rowlands, 2020). The same applies when transposing the accuracy of the camera system to distant objects to locally obtain the corresponding spatial resolution of the system.

For an impact close to the center of the video frame, the simplified projection of the circular area perpendicular to the camera viewpoint generates a right circular cone with an aperture ($\theta_a$ [°]) given by the adjusted diagonal field of view (FOV) of the lens objective for the cropped portion of the camera sensor used at the desired video resolution ($Res_{horizontal} \times Res_{vertical}$ [px]) using Eq. (18) as follows:

$$\theta_a = 2\tan^{-1} \frac{p_r \tan(0.5FOV)}{0.5\sqrt{Res_{horizontal}^2 + Res_{vertical}^2}} \; , \tag{18}$$

The intersection of the cone with the terrain forms a conic section in the shape of an ellipse. The semimajor axis ($a$) of this ellipse of accuracy corresponds to the orientation with the lowest precision and accuracy from the camera viewpoint (i.e., in the direction of the "steepest" depth gradient), and the semiminor axis ($b$) corresponds to the maximal precision and accuracy. The angle ($\varphi_{ter\text{-}cam}$) of the terrain with the camera viewpoint, and thus, the angle of the conic section with the central axis of the cone is given by Eq. (19) as follows:

$$\varphi_{ter-cam} = sin^{-1} \frac{\left|\overrightarrow{\Delta X_{ter-cam}} \cdot \vec{N}\right|}{\left\|\overrightarrow{\Delta X_{ter-cam}}\right\|\left\|\vec{N}\right\|} \; , \tag{19}$$

The greater the viewpoint is perpendicular to the terrain, the less the ellipse of accuracy is elongated while the semiminor axis remains constant. The semimajor axis is reduced to the shortest length, and it is equal to the semiminor axis for the special case where the terrain is perpendicular to the viewpoint (e.g., the circular yellow ellipse from UAV camera 1 in Fig. 3). The constant semiminor Axis $b$ length, corresponding to the maximal precision and accuracy, is given by Eq. (20) as follows:

$$b = \left\|\overrightarrow{\Delta X_{ter-cam}}\right\| \cdot \tan\frac{\theta_a}{2} \; , \tag{20}$$

The minimal precision and accuracy from one viewpoint can be found from the length of the semimajor Axis $a$ using Eq. (21) as follows:

$$a = \frac{\left\|\overrightarrow{\Delta X_{ter-cam}}\right\| \sin^{\theta_a}/_2}{\sin\left(\varphi_{ter-cam} - \theta_a/_2\right)} \; , \tag{21}$$

For situations where the viewing angle is far from being perpendicular to the terrain, the ellipse of accuracy is very elongated. An impact position picked in such a configuration would be greatly inaccurate in the direction of the "steepest" depth gradient. This situation can be greatly improved if the impact is also captured from a second point of view. Indeed, two or more viewpoints can be combined to maximize the precision and accuracy to the constrained area of the overlapping ellipses. In doing so, the accuracy can approach that given by the semiminor axes, even if the accuracy ellipses are elongated, as shown by the green areas in Fig. 3 and Fig. 4.

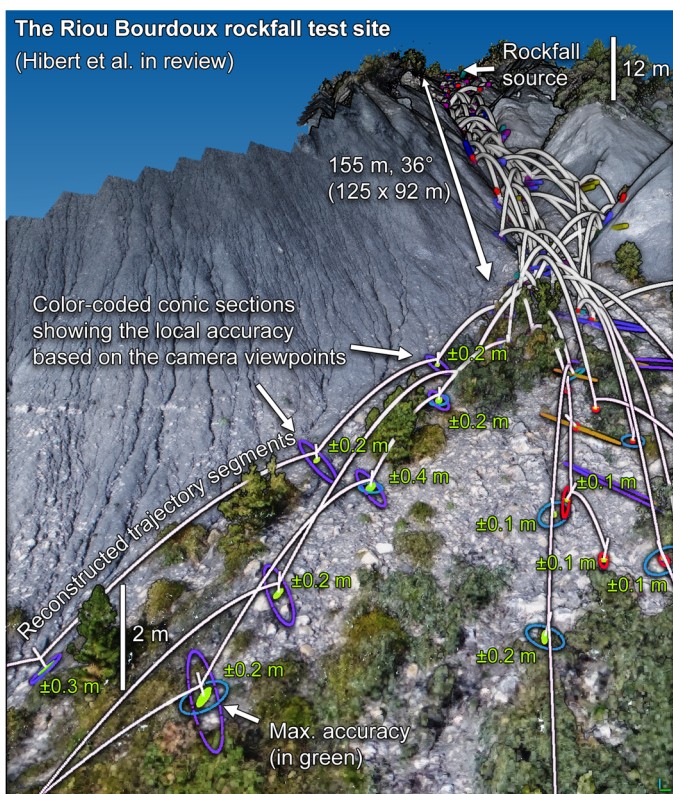

**Figure 4: Examples of spatial accuracies from the projection of ±4 px picking accuracies for the Riou Bourdoux rockfall test site that involved multiple ground based and airborne camera viewpoints (Hibert et al. in review). The maximized local precision and accuracy of the 376 constrained areas of the overlapping ellipses are shown in bright green like in the previous figure. The maximal sizes (worst values) of some maximized local accuracies are written in bright green next to their impacts. Note that each impact accuracy is unique and depends on many variables like the impacted terrain geometric configuration and texture, its distance from the viewpoints and the video acquisition setup that influence locally the ease of visually identifying the impact location.**

### 3.1.2 Acquisition setup

Concerning the sharpness associated with the level of detail of the footage, counterintuitively, lower resolution footage with a narrow FOV can be better than ultra-high resolution for this method. Sharp and detailed ultrahigh-resolution footage (e.g., at 8K UHD resolution, 7680 × 4320 pixels) requires the use of a proper large sensor and a high-end lenses combo designed to have enough resolving power and acutance for that task. For example, a 4K UHD (3840 x 2160 pixels) Super 35 sized sensor with an effective area 24.89 mm wide is ~154 pixels wide per millimeter. It has a corresponding capacity to resolve a

maximum of 77 line pairs at its Nyquist frequency, i.e., when half of the 154 line pairs match the corresponding 154 pixel binning sampling (vertical axes of Fig. 5). Lenses are often the limiter at high resolution given their lower contrast at such high line pairs per millimeter.

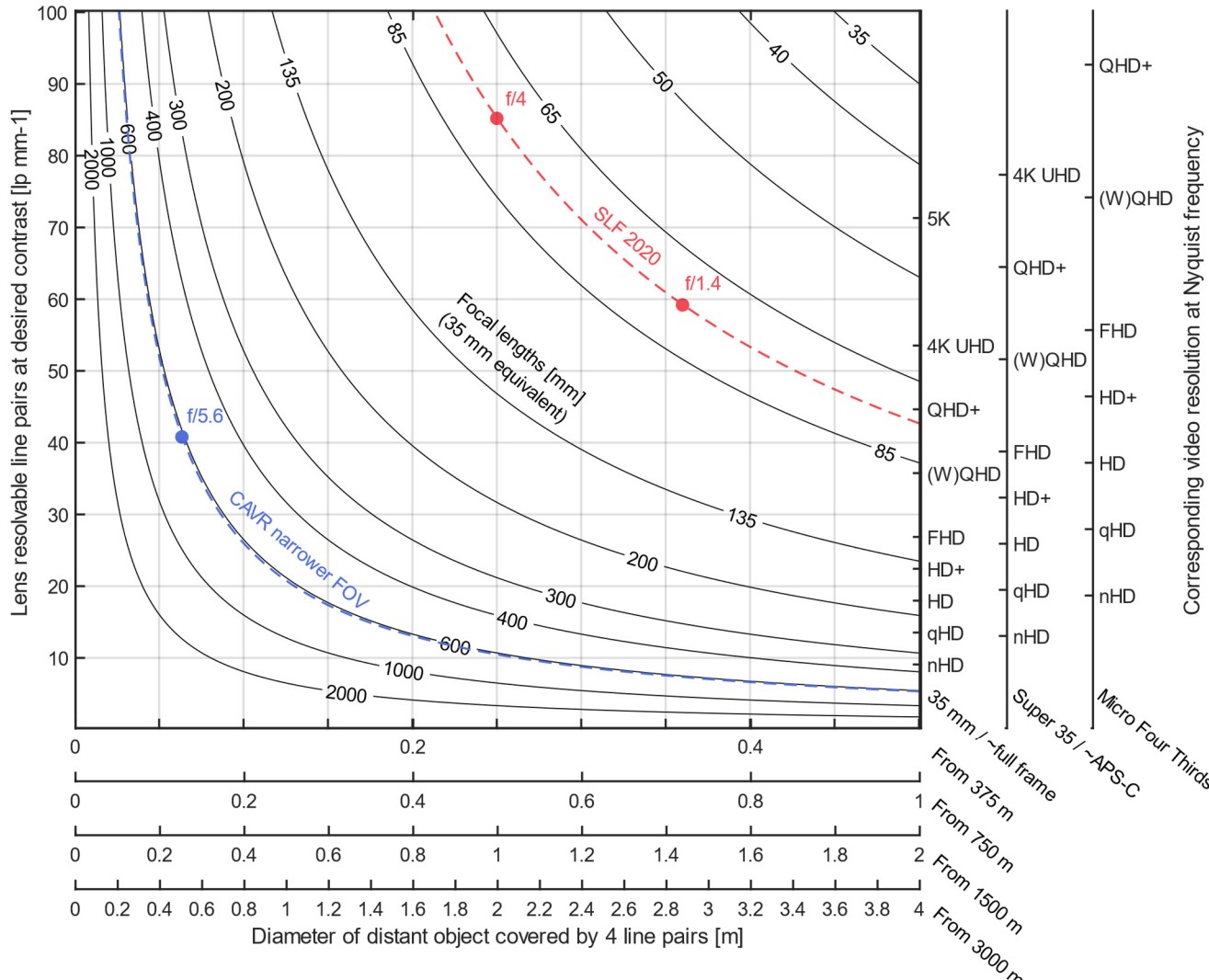

**Figure 5: Abacus putting in relation to the lens resolvable line pairs given by the modulation transfer function (MTF) to the corresponding video resolution at maximum resolving capacity for different common sensor sizes and perpendicular distant object diameters equivalent to 4 line pairs that can be resolved through lenses of different fields of view (FOVs) given by their 35 mm equivalent focal lengths. The equivalent focal lengths of the Canon at 400 mm and Zeiss 55 mm lenses used on cameras with Super 35 / ~APS-C sensor sizes for the Chant Sura rockfall test site used as an example are shown with blue and red dashed lines. The central MTF values at 50% contrast for the lenses are shown with dots for different aperture values. Unless capturing objects with fine grid patterns where aliasing could be a problem, it is recommended to sample above the Nyquist frequency to avoid being constrained by the camera body/sensor before downscaling to the resolvable video resolution. For example, the Samsung NX1 camera body used for the CAVR method samples at ~6.5K from the whole ~APS-C sensor readout before performing the in-body downscaling).**

Combining the previous concept of the lens-sensor resolving power with the precision and accuracy concepts of the previous section, the on-sensor resolving power can be transposed to a perpendicular distant object to predict the level of details that can be captured in the center of the frame for a desired contrast (Fig. 5). As shown, the amount of captured detail at a lower resolution (e.g., at HD or FHD) with a narrower FOV can be very similar or even better than if captured at an ultra-high resolution with a wider FOV (Fig. 5). In the following paragraphs, the previous data captured at the Chant Sura rockfall test

site with a wider FOV and steady cameras (Caviezel et al. 2020), referred to with the acronym "SLF 2020", are compared to newer footage captured with a narrower FOV and manual tracking allowed by the CAVR method as acquisition examples (Fig. 5).

    The WSL Institute for Snow and Avalanche Research SLF performed novel rockfall experiments with instrumented rocks at the Chant Sura test site (Fig. 6) sometime involving a 5 m tall by 60 m long 2000 kJ ROCCO flexible barrier from Geobrugg

(Caviezel et al. 2019, 2020, 2021; Sanchez and Caviezel, 2020). They opened their experiment to the public with Geobrugg at the GEO-summit 2019 conference and publicly shared part of the acquired data in Caviezel et al. (2020). The transparency of such an action toward open science should be emphasized. The accessible data can be very helpful to the geohazard community when assessing the sensitivity of current rockfall simulation software and for finding the right simulation parameters to be used for similar sites.


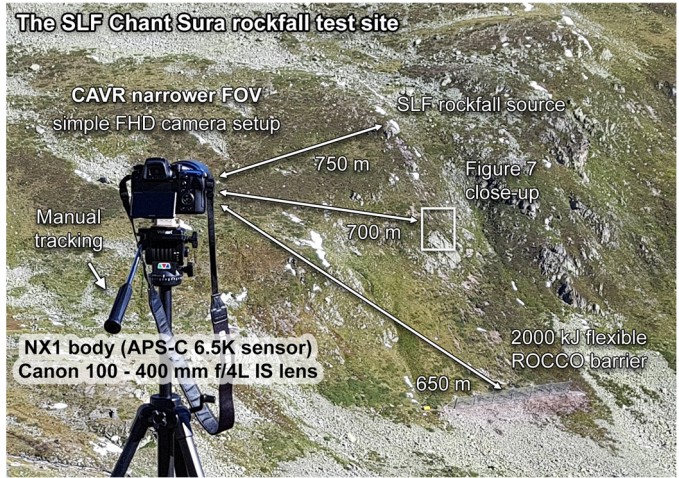

**Figure 6: Simple camera setup based on the presented acquisition concepts for the CAVR method tested here at the Chant Sura rockfall test site during the September 13th 2019 SLF experiment. The white rectangle shows the close-up area of the site used in Fig. 7 to compare the SLF RED video footage to the FHD footage from this simple camera setup. The Samsung NX1 camera body**

**from 2014 could deliver 4K UHD footage (6.5K full sensor readout downscaled to 4K), but the lens from 1998 used here does not have the resolving power for the resolution on the ~super 35 / APS-C sized sensor (23.5 x 15.7 mm, crop factor of 1.53). Thus, the footage was recorded at FHD and could reach a higher frame rate instead (119.88 fps).**

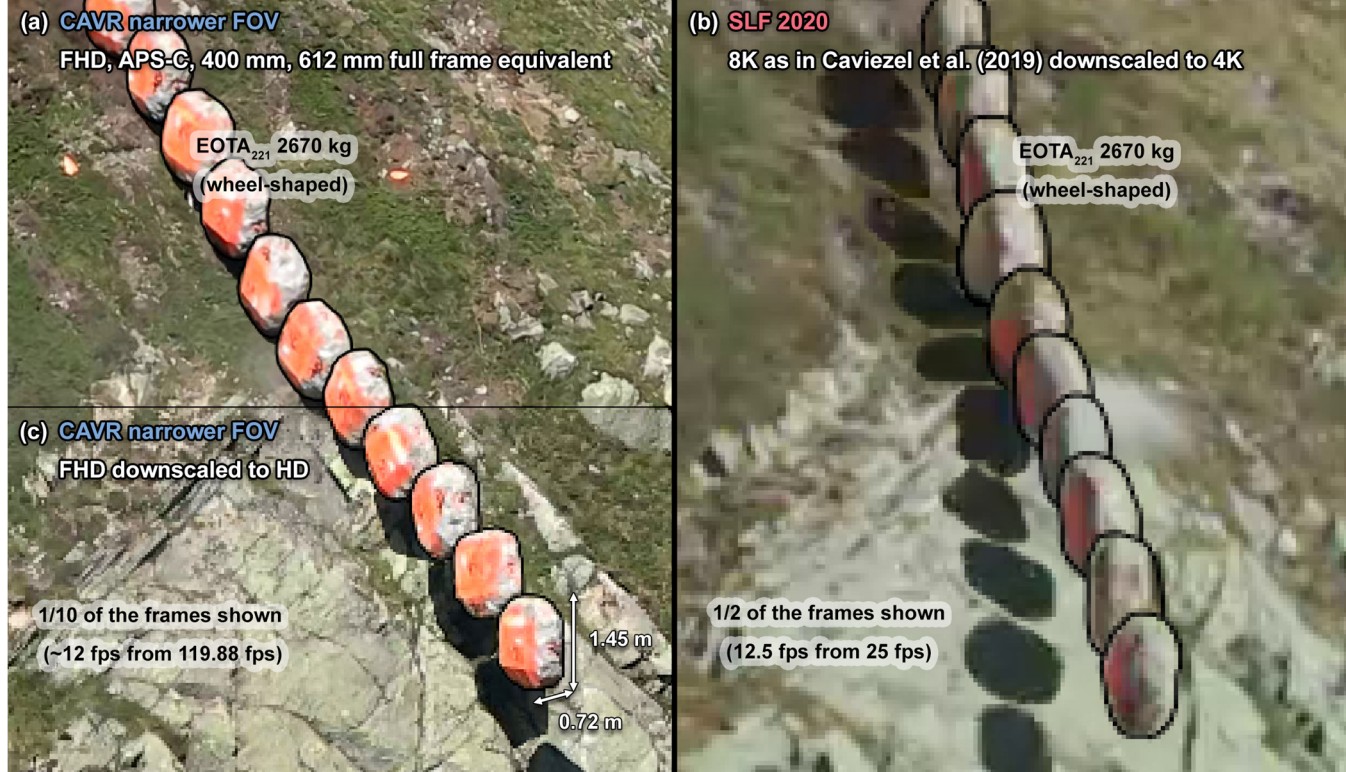

**Figure 7: Comparison of the FHD (a) and HD (c) narrower FOV video footage captured with the CAVR camera setup shown in Fig. 6 with the wider SLF RED video footage (b) from Caviezel et al. (2020), described in Caviezel et al. (2019) for rocks with similar trajectories. The visible outline of the rocks is highlighted with a black line roughly hand drawn on the stacked frames to help distinguish the rocks from their background. Note the difference in sharpness and the level of details of the footage related to the captured resolutions and the field of view. For scale, the 1 m³ 2670 kg rocks have their rotations aligned around their shortest**
**diameters of 0.72 m and have their longest diameters equal to 1.45 m. They are the largest blocks tested at that site in Caviezel et al. (2021), and therefore, are the easiest to see on the video footage. The smallest equant blocks of 45 kg are approximately one-fifth of the length of these blocks and require detailed footage for proper tracking.**

In parallel to the common acquisition setup previously used by the SLF at that site (Caviezel et al., 2019), the alternative CAVR camera setup following the previously described concepts was deployed from one viewpoint for the rockfall

experiment performed on September 13th, 2019 (some footage is publicly available in Caviezel et al., 2020). The simple and affordable alternative setup consists of a camera capturing at FHD resolution, 119.88 fps and a fast shutter speed, coupled to a zoom telephoto lens used at 400 mm (~612 mm full frame equivalent) for a narrow FOV of approximately 4° and manually panned to track the rock projectiles (Fig. 6).

The CAVR narrower FOV FHD footage is compared to the SLF 2020 older 8K UHD high-end RED video footage from the

SLF (Caviezel et al., 2020) in Fig. 7 for rocks sharing similar trajectories. The resolution of the CAVR narrower FOV footage is also downscaled to HD in Fig. 7.c to compare with the SLF 2020 footage at 4K UHD downscaled from 8K UHD visible in Fig. 7.b. Even with a resolution reduced by half from FHD, it is possible to see that the sharpness of the CAVR narrower FOV footage at HD surpasses the SLF 2020 high-end RED footage. Indeed, the CAVR narrower FOV footage

(Fig. 7a) resolves smaller distant objects, as foreseen in the abacus (Fig. 5) and confirmed by the sharper edges and the visible details around the bright outcrops. The CAVR narrower FOV footage following the previously described concepts shows more details and sharpness thanks to the narrow FOV used, despite having been captured with a five-year-old camera body coupled with a twenty-year-old telephoto lens at the time of performing the experiment.

The limitations by the lens, as previously observed on the SLF 2020 older footage, are likely to occur on super 35 / APS-C sensors, even with the extremely sharp Zeiss Otus 55 mm f/1.4 used in Caviezel et al. (2019). Reducing the aperture could help (Fig. 5), as no lens is perfect when wide open, especially in the corners. This is especially true when the camera is kept still with the rocks moving across the frame, potentially reaching the corners. However, with the CAVR reconstruction method, one can pan to track the rock projectiles to keep them close to the center of the frame where lenses are most of the time at their best, allowing a wider aperture to be used without degrading the sharpness in the center. A wider aperture comes with a shorter exposure period from a faster shutter speed or a lower ISO sensitivity, reducing the motion blur or the noise level of the captured footage. With the panning motion, a narrower FOV can be used while tracking the moving rock, as with the FHD camera setup shown in Fig. 6. A camera body with a fast sensor read should be used to reduce the rolling shutter skew distortion with such configuration.

Therefore, as conceptualized in Fig. 3 and Fig. 5 and shown in Fig. 7, more detail around the impact points can be obtained at lower resolution if the panning motion and the narrow FOV are combined for tracking the rocks. This in turn allows a higher constant frame rate and a fast shutter speed to be used, as often required for tracking the angular velocities and for timing the impacts. Lower resolution file handling and playback are also simplified because the footage can be played fluently and edited efficiently on most common computers to add, for example, an overlaying timecode and electronic image stabilization. Additionally, blurry footage can still be sufficient for timing impacts if they can be located from the impact marks left on the terrain. As the 3D detailed terrain model is the spatial reference for the impact positions and not the cameras, this CAVR method is flexible to many types of video footage available. Thus, it is well suited for reconstructing 3D trajectories from nonoptimal previous records to help gather data needed for site-specific calibration of sensitive rockfall simulations. Consequently, valuable rockfall data, such as the data gathered by the SLF (Caviezel et al., 2021), could also be acquired with an affordable camera setup and from previously witnessed rockfall events.

## 3.2 Digital terrain model

A corresponding detailed 3D model of the terrain is needed to extract the coordinates of the impacts for reconstructing the trajectories. As covered in Noël et al. (2021), it can be acquired in many ways, e.g., by Structure from Motion photogrammetry (SfM), by airborne, mobile, or terrestrial laser scanner (ALS, TLS). The SfM method preferable because it is often exempt of occluded part and properly capture the terrain roughness as perceived by the rocks (Noël et al., 2021). It can also texture the 3D model from the acquired pictures, which is very helpful to visually locate the impacts and extract their position coordinates when no indentation mark or scar is visible. Other methods can be textured from projected photos and orthophotos or from the return signal's intensity. Vegetation is often not a problem for freshly affected sites, since large

rockfall events usually remove part of it. Artifacts and bushes might be present, however, and should be avoided when evaluating the impact position and the local terrain orientation. They can be highlighted by artificial shading methods, such as the eye dome lighting method (EDL) (Boucheny, 2009) or the ambient occlusion method (PCV) (Duguet and Girardeau-Montaut, 2004; Tarini et al., 2006). Local geomorphological features and impact marks are also highlighted with these shading methods (Fig. 8). These methods can also be combined with coloring methods based on the local terrain orientation (e.g., the Coltop method by Jaboyedoff et al. (2007), which is also implemented with a slightly different color distribution in the CloudCompare open-source software (Girardeau-Montaut, 2006)). A comparison of two terrain models, from before and after the rockfall event(s), can also help highlight the impact marks if such models are available, as shown by Caviezel et al. (2019).

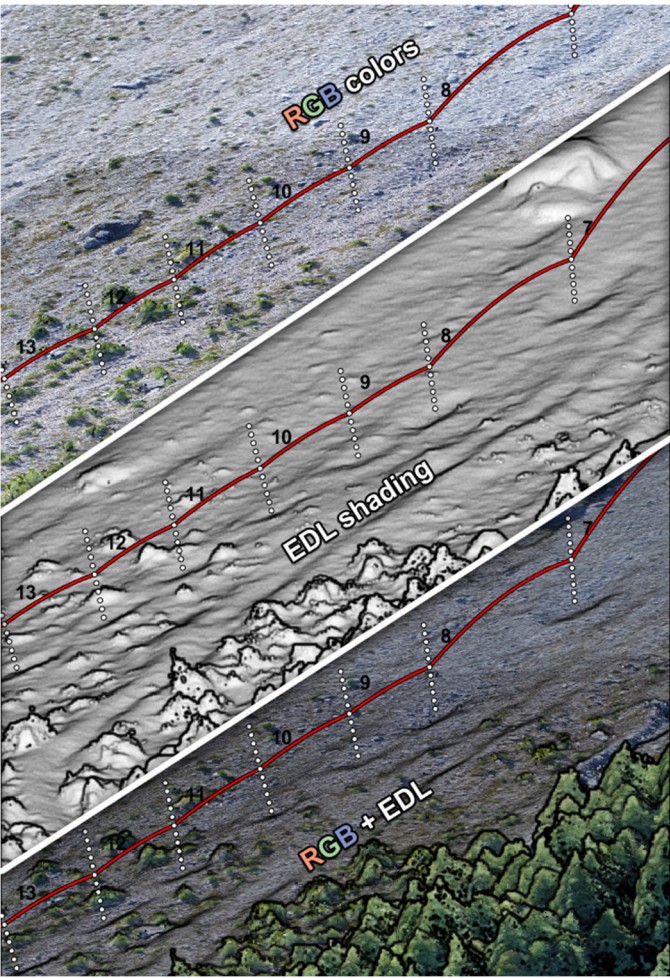

**Figure 8: Same 3D scene with different rendering settings of the 3D terrain model. The EDL shading filter can be very useful for highlighting artifacts and impact marks. The scenes are rendered in a custom tool developed to help assist the reconstruction process, as later described. The trajectory under reconstruction shown in red with white dotted normal vectors from each impact mark visible on the ground corresponds to the block with the longest runout from the 2015 Mel de la Niva rockfall event (Noël et al., in preparation; Lu et al., 2018). It transitions from longer freefalling phases to a "rolling-bouncing" phase.**

### 3.3 Rock block geometry

It is then necessary to evaluate the rock block geometry to properly offset the impact positions to the center of mass. It can be tempting to simply use the impact positions without the offsets, but the resulting trajectories would not have the right lengths, which is highlighted by Volkwein et al. (2011) as shown in Fig. 1, and wrong reconstructed velocities would be obtained (see Appendix B about how the change in impact-to-impact distance can affect the results). The rock geometry can be evaluated from on-field measurements or with 3D models acquired by SfM or by mobile and TLS methods. The mass can be determined from the volume ($V$) of the rock and the volumetric mass density ($\rho_{rock}$) of rock samples (assuming a homogeneous distribution of the mass). First, an estimation of the volume can be done by simplifying their shapes to ellipsoids from measuring the $d_1$, $d_2$, and $d_3$ diameters of the rocks on field, with $d_1$, $d_2$ and $d_3$ being the lengths of the longest, intermediate, and shortest sides, respectively. In that case, the geometric properties are given with the Eq. (22) to Eq. (26) as follows:

$$V = \frac{\pi}{6} d_1 d_2 d_3 , \tag{22}$$

$$m = \rho_{rock} V , \tag{23}$$

$$I_{d3} = \frac{1}{20} m \left( d_1{}^2 + d_2{}^2 \right) , \tag{24}$$

$$I_{d2} = \frac{1}{20} m \left( d_1{}^2 + d_3{}^2 \right) , \tag{25}$$

$$I_{d1} = \frac{1}{20} m \left( d_2{}^2 + d_3{}^2 \right) , \tag{26}$$

where $I_{d1}$, $I_{d2}$, and $I_{d3}$ are the principal moment of inertia when the rock is rotating around the $d_1$, $d_2$, and $d_3$ axes respectively. Such estimation from ellipsoids can be performed efficiently from 3D models containing numerous rocks using the automatic method described in Steer et al. (in review). The precise moments and principal axes of inertia can also be identified from an analysis on the 3D meshed model of individualized rocks (e.g., with MeshLab from Cignoni et al. (2008)). The dimensions can then be expressed based on the size of the bounding boxes aligned on the main inertia axes of the rock. This is similar to the adjusted bounding box method presented by Bonneau et al. (2019); therefore, the 3D meshed model should be exempt from artifacts. Then, the offset distance for each impact can be approximated to half of approximately 90% of either $d_1$, $d_2$, $d_3$ or in between based on the visually identified amount of scarring and rock configuration at impact. It is then possible to extrapolate the position of the center of mass from the coordinate of the center of each impact mark on the ground with the proper offset normal to the terrain. Trajectories with the rock's translational velocity can be reconstructed from these positions and the impact time using Eq. (5) and (6).

If the video frame rate is sufficient, the angular velocity can be estimated by counting the number of rock rotations completed over the freefalling period between the impacts. The main axis that the rock rotates around should be noted. The angular momentum and kinetic energy can then be estimated by selecting the corresponding moment of inertia. As this is time-consuming, the method can be combined with the approaches from Volkwein and Klette (2014) and Caviezel et al. (2019) to retrieve the angular velocity from inboard gyroscopes.

### 3.4 Visual validation and fine-tuning of the results

It is recommended to validate visually if the reconstructed trajectory matches what is seen from the video footage. For that, the trajectory can be loaded in a 3D visualization software together with the terrain model. They can then be analyzed visually by placing the viewpoint from the same position as the cameras used to capture the event with perspective and a similar field of view. Properly reconstructed trajectories should be aligned on the rocks from the video footage. In other words, they should align with the same background elements on the terrain model (e.g., characteristic ground textures,

bushes, or rocks on the ground) as those momentarily occluded by the falling rocks in the video footage when the rocks pass in front of them. This approach is later used to compare the older reconstructed trajectories from the SLF (Caviezel et al., 2020) with newer trajectories using the CAVR reconstruction method.

The total apparent kinematic coefficient of restitution ($COR_v$) should be under 1.00. Exceptionally, it can be slightly above 1.00 if a considerable amount of angular momentum is transferred to a translational momentum at the impact. Overall, the

430 total kinematic energy cannot be greater after the impact point than before. If the opposite is observed, the reconstructed impact must be fine-tuned or discarded for impacts with long rock-ground interactions.

### 4 Computer-assisted reconstruction

To facilitate and homogenize the reconstruction process, we develop a tool with a graphical user interface (GUI) that incorporates the previously mentioned concepts of the reconstruction method. The 3D detailed terrain model can be

visualized with a perspective from two viewpoints simultaneously (Fig. 9). The field of view can be adjusted to match the video footage. The prerendered terrain from the chosen viewpoints can be shown with its textured RGB colors only, with only the EDL shader, or a combination of the two to facilitate the localization of the impact point and the eventual scars (Fig. 8). The terrain can be explored by panning around the camera point of view on either one of the two viewing windows, reproducing the panning motion from the video footage to track the rock projectiles. The other window then pans

automatically to follow the same part of the terrain tracked in the center.

From there, a trajectory and impact number to be edited must be selected. The frame number at which the impact occurs in the main video file can be set. The impact time is then calculated from the constant frame rate set for the video file (e.g., the 13[th] impact being reconstructed in Fig. 9 has an impact time related to the beginning of the cropped video file of 1802/119.88 [fps] = 15.03 s). This is quicker than having to enter the time in minutes, seconds, and frame, and reduces the risk of

445 transcription error. The diameter of the rock to use for the desired offset must also be set (e.g., ~90% of $d_1$, $d_2$, $d_3$ or in between, depending on the observed amount of scarring and the visually identified rock configuration at the impact). Half of this value is then used for the offset to place the center of mass using the impact detection algorithm described in Noël et al. (2021).

It is possible to define an impact position on the ground by either directly pointing at the terrain model with the mouse cursor

or by entering the coordinates manually. The normal to the terrain is then updated automatically in real-time using the

efficient impact detection algorithm that works on a detailed 3D terrain model while considering the rock size (Noël et al., 2021). The normal is shown as a white line perpendicular to the terrain in the footprint of the rock and follows the mouse cursor if the position is defined with it. The trajectory is then updated in real-time, properly offset perpendicularly to the terrain from the mouse cursor. The impact time can also be slightly adjusted by scrolling while picking the impact point with the mouse to see the effect in real-time on the reconstructed parabolas.

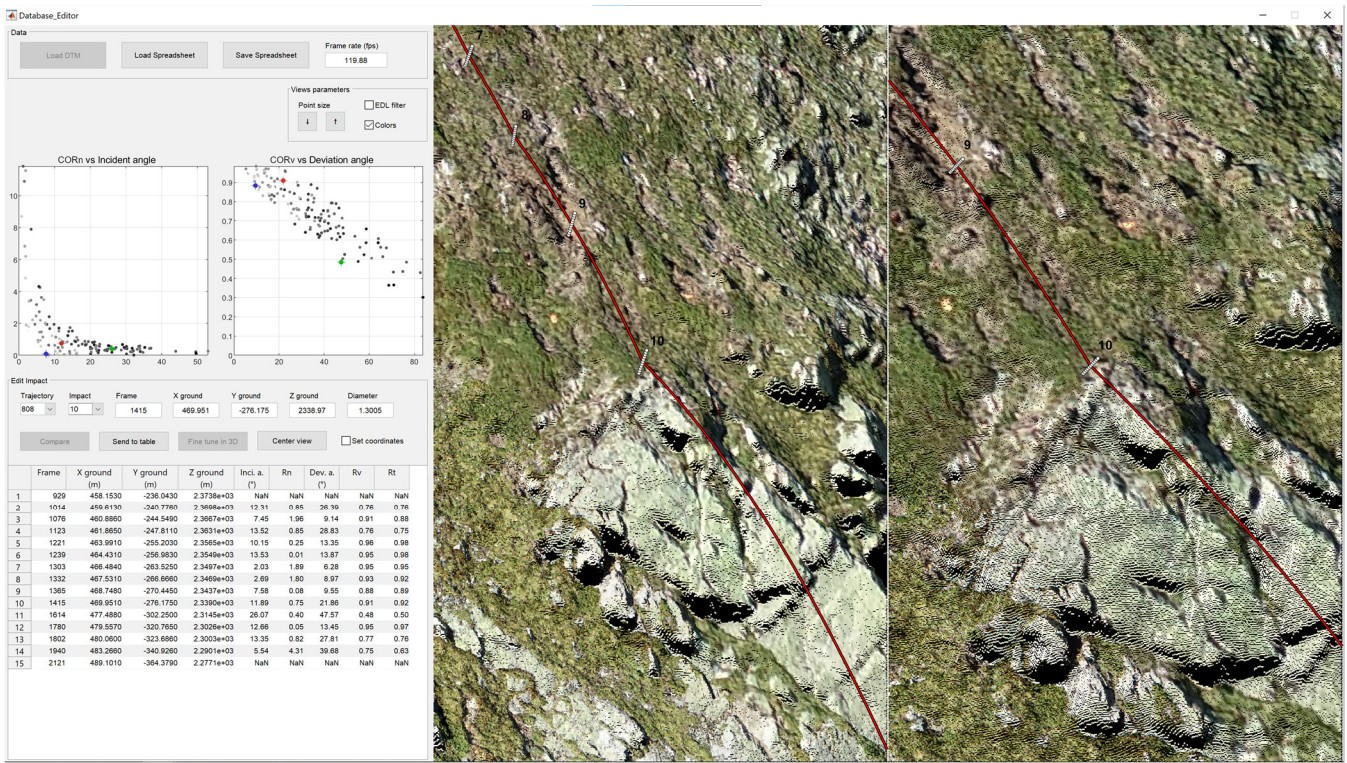

**Figure 9: Graphical user interface for assisting the reconstruction process, shown here during the reconstruction of the trajectory of the 8[th] rockfall run performed by the SLF at the Chant Sura test site on September 13[th], 2019 (Caviezel et al., 2020; Sanchez and Caviezel, 2020). A trajectory being reconstructed is shown in red in the two viewing windows, with the impact points and normal vectors for the automatic offset shown in white. The on-ground impact position can be entered manually (minus a global shift translation to bring the coordinates close to a local origin) or set by clicking directly on the 3D terrain model. The reconstructed trajectory is updated in real-time following the mouse cursor.**

This emphasis on the real-time updating of the reconstructed trajectories is important because the validation and fine-tuning processes then become part of the reconstruction process. The goal, after all, for this process is to reconstruct data that match as much as possible with what is observed, while sometimes having to struggle with some unknowns or nonoptimal video footage. Therefore, having the flexibility to instantly see the reconstructed result when hesitating in between two frames for the impact time or when hesitating on the impact location by a few centimeters to decimeters truly helps find the best parameters that make the trajectory match what is seen on the video footage. Of course, the quality of the reconstructed trajectories depends on the quality of the input footage and the 3D terrain model. Therefore, if it is impossible to see part of a

trajectory and its bounding impacts on the footage or the terrain; then this part should simply be discarded or kept for qualitative purposes only.

To further ease the reconstruction and validation process, some reconstructed properties of the impacts are shown in two graphs ($COR_N$ as a function of $\theta_I$ and $COR_v$ as a function of $\theta_{dev}$). They are also updated in real-time. The edited impact and its direct neighbors are highlighted with different colors, with red for the current impact, blue for the previous impact, and green for the next impact, if present. The other impacts are shown in different shades of gray, with darker values if the period preceding and following the impact are longer, and therefore, with relatively more precise reconstructed values. It is possible to quickly notice if an impact is behaving strangely, such as, for example, if the total returned translational velocity is largely above the incident one or if the impact is not following the trends. From there, more emphasis can be spent on understanding the reasons behind the strange behavior of an outlier (e.g., the rock shattered) and fix the impact if any error is made on the position or timing. However, this should never be used to choose parameters that would force a fit on the trends. See Appendix A about the significance of air drag if considering using the CAVR method with small rock projectiles and/or important freefalling distances reaching high velocities. More details about the positioning and timing precision of the method can be found in Appendix B.

## 5 Method comparison

In this section, the presented trajectory reconstruction method (CAVR) is challenged when used next to the results of an existing peer reviewed method. A comparison of the produced results can provide a validation that the presented method produces valid results. The concept of this comparison is as follows: if the rockfall trajectories are properly reconstructed, they should align with the real rock positions from the video footage. The reconstructed trajectories and energies should also correspond with those from the existing reconstruction method. Such a comparison would show that the presented flexible rockfall reconstruction method reproduces proper 3D trajectories from real rockfall events or experiments.

### 5.1 Comparison approach

For this exercise, the CAVR method is used with the presented computer assisting tool to reconstruct the 9 trajectories from the SLF rockfall experiment performed on September 13[th] of 2019 at the Chant Sura site (Sanchez and Caviezel, 2020). The reconstruction is quickly performed in approximately one day for the purpose of this comparison, and with the nonoptimal configuration of using only one viewpoint. The 119.88 fps FHD video footage with a narrower FOV using the camera setup previously described (Fig. 6) is used for the reconstruction (footage available for the 6[th] and 7[th] rockfall runs in Caviezel et al., 2020).

The detailed digital terrain model (DTM) used as a spatial reference for the reconstruction corresponds to the model from before the experiment performed that day. The DTM is generated by the SLF with structure from motion photogrammetry using precisely geolocated pictures acquired with a DJI Phantom 4 RTK. For the reconstruction with the CAVR method, the

terrain model is textured based on the orthophoto after the experiment using the publicly available terrain models and orthophotos from the SLF in Caviezel et al. (2020).

With one camera input per rock publicly available for that site at the time of writing these lines, it is not possible to independently reproduce Caviezel, et al.'s (2019) method, which relies on video stereo pairs. Therefore, the method comparison focuses on comparing the reconstructed trajectories from the CAVR method with the rocks visible on the stacked aligned FHD video frames. Nevertheless, the older 3D reconstructed trajectories from the SLF (Caviezel et al., 2020) based on Caviezel et al. (2019) can be visually compared side-by-side with the newer trajectories based on the CAVR method. The newer reconstructed trajectories, however, differ by being offset to the center of mass of the rocks instead of being reconstructed directly from the contact points on the ground, as illustrated in Fig. 1. This side-by-side height difference affects the reconstructed bounce heights by approximately half of $d_1$.

The intercepting reconstructed impacts at the 2000 kJ flexible ROCCO barrier from Geobrugg in Sanchez and Caviezel (2020) are used to refine the comparison with the reconstructed energies and bounce heights. Apart from the fact that the intercepted rocks were all stopped by the flexible barrier, the related information is here kept succinct, as the authors do not want to impinge on future publications focusing on the behavior of the flexible barrier. The reader is referred to Sanchez and Caviezel (2020) for more information about the novel experimental setup with the flexible barrier.

## 5.2 Results and discussion

The reconstructed translational velocities and related energies, parabola lengths, and vaulted shapes from the two side-by-side 3D reconstructed trajectories (r1 and r2) overlaying the stacked video frames of the 6th and 7th rockfall runs in Fig. 10 are similar. Despite being visually aligned with the center of the orange rocks most of the time, slight rare misalignments persist between the CAVR reconstructed trajectories and the visible rocks in the video footage (Fig. 10a, Fig. 10c and Fig. 11). This suggests that the reconstruction could be refined further, especially if improved by also using video footage from other view angles. Visual alignment mismatch is also present on the SLF's older reconstructed trajectories (Fig. 10b and Fig. 10d).

Unlike in Caviezel, et al. (2019, 2021), only dissipative impact processes are obtained with the CAVR approach. Indeed, no apparent gain of kinetic energy at impact that would be manifested by $COR_v$ above 1.00 is obtained. This is because the presented reconstruction methodology excels in the resolution of smaller bounces in complex and steep impact configurations, especially with the easy validation process from the real-time update of the reconstructed trajectories with the computer-assisted approach. The CAVR method is, however, limited to short rock-ground interactions simplified to single impact points offset from the ground. Longer rock-ground interactions in steep terrains have different potential energies at the start compared to the end of the interactions that contribute to some of the apparent gain of kinetic energy previously obtained by the SLF (Caviezel et al., 2019).

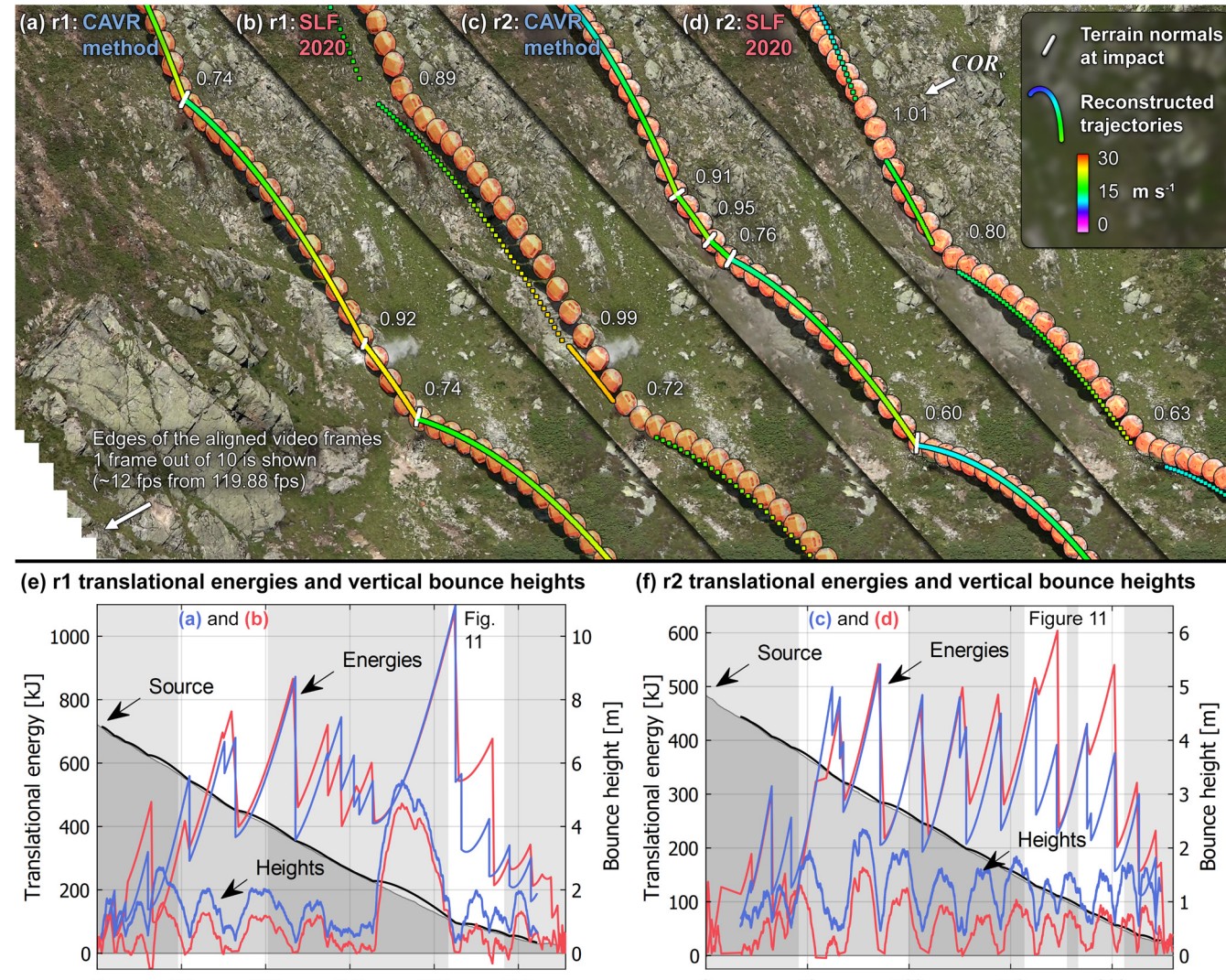

**Figure 10: Side-by-side comparison of the reconstructed translational velocities, positions, translational energies, and bounce heights of the reconstructed trajectories with the CAVR method compared to the older trajectories from the SLF (Caviezel et al., 2020). The r1 and r2 rockfall runs correspond to the 6th and 7th runs performed by the SLF on September 13th, 2019. The 3D reconstructed trajectories are overlayed on the stacked frames showing the rock positions every 10 frames (every ~1/12th of a second for the footage captured at 119.88 fps with the camera setup shown in Fig. 6). For scale, the dimensions of the 2670 kg orange reinforced disk or wheel shaped concrete rock are 1.45 m ($d_1$ and $d_2$) by 0.72 m ($d_3$). The reconstructed trajectory segments are shown in black with their respective slope profiles in gray as background elements of the 2D vertical profiles shown in (e) and (f). The characteristic sawtooth shape of the energy profiles helps distinguish them from the bounce heights. The values from the CAVR method are shown in blue while those from the SLF are shown in red for a quantitative comparison. The portion of reconstructed trajectory segments shown over stacked video frames in (a), (b), (c), (d) and in Fig. (11) are highlighted in the 2D profiles (e) and (f) with white vertical bands.**

The bounce heights from the center of mass of the rocks with the CAVR method are always above the terrain surface and have values that rarely fall under one radius of the rocks ($d_1$ of 1.45 m) (Fig. 10e and Fig. 10f). They follow those from the

older reconstructed trajectories from the SLF, but they are slightly higher by the equivalent of approximately one radius of the rocks, and they are never negative. This is due to the applied offset perpendicular to the impacted terrain bringing the reconstructed trajectories to the center of mass of the rocks for each impact.

Most energy peaks from the typical saw-tooth rockfall energy profiles align and reach similar values between the two methods. This shows that the presented flexible rockfall reconstruction method can reproduce proper 3D trajectories from real rockfall events or experiments. Focusing on the few abnormal local differences, the reconstructed translational energy values mostly differ for r1 (Fig. 10e) for one freefalling phase after reaching the maximum translational energy from 216 to 233 m. For r2 (Fig. 10f), they mostly only differ for two freefalling phases from 163 to 173 m and from 190 to 201 m. These rare abnormal energy mismatches deserve a closer look.

The reconstructed trajectory segments of the few abnormal energy mismatches previously highlighted are detailed in Fig. 11. For proper trajectory reconstruction, the impact position and timing must be chosen precisely by deciding on the right freefalling period (see Appendix B about the positioning and timing precision). The timed dashed pattern should follow the appearance of each new stacked frame if the timing of the chosen period is correct. The chosen bounding impact position for the beginning and the end of the three freefalling reconstructed parabolas should also align with the yellow frames with thicker added black contours corresponding to the observed rock positions at the start and the end of each period. With the timing and position of the bounding points of the parabola matching the observations, the reconstructed parabolas from the CAVR method in Fig. 11a, Fig. 11c and Fig. 11e align well visually with the observed positions of the freefalling rocks. Therefore, the resulting reconstructed translational velocity and energy values are close to reality. Sharp detailed video footage with a high frame rate for a precise time resolution following the presented acquisition concepts and combined with the computer-assisted tool helps find the right freefalling period and transpose the impact positions with accuracy. In the 3D space, the r1 parabola reconstructed with the CAVR method in Fig. 11a is 2.21 m away in average from the one from the SLF (Fig. 11b), with a S.D. of 0.89 m. Concerning the two r2 parabolas in Fig. 11c and Fig. 11e, they are respectively separated by 0.77 m, S.D. of 0.10 m and 0.79 m, S.D. of 0.02 m from the SLF parabolas (Fig. 11d and Fig. 11f).

Conversely, choosing bounding impact positions further apart or freefalling periods shorter than the observed would artificially boost the reconstructed translational velocities as longer travel distances must be connected in shorter periods. This can explain the three abnormal higher mismatching reconstructed translational energies of the older reconstructed trajectories from the SLF shown in Fig. 10e and Fig. 10f and detailed in Fig. 11b, Fig. 11d and Fig. 11f. Such timing and positioning imprecision can also contribute to some of the apparent gain of kinetic energy at impact manifested by $COR_v$ above 1.00 or the positive energy ratio obtained by the SLF with other reconstruction methods in Caviezel, et al. (2019, 2021).

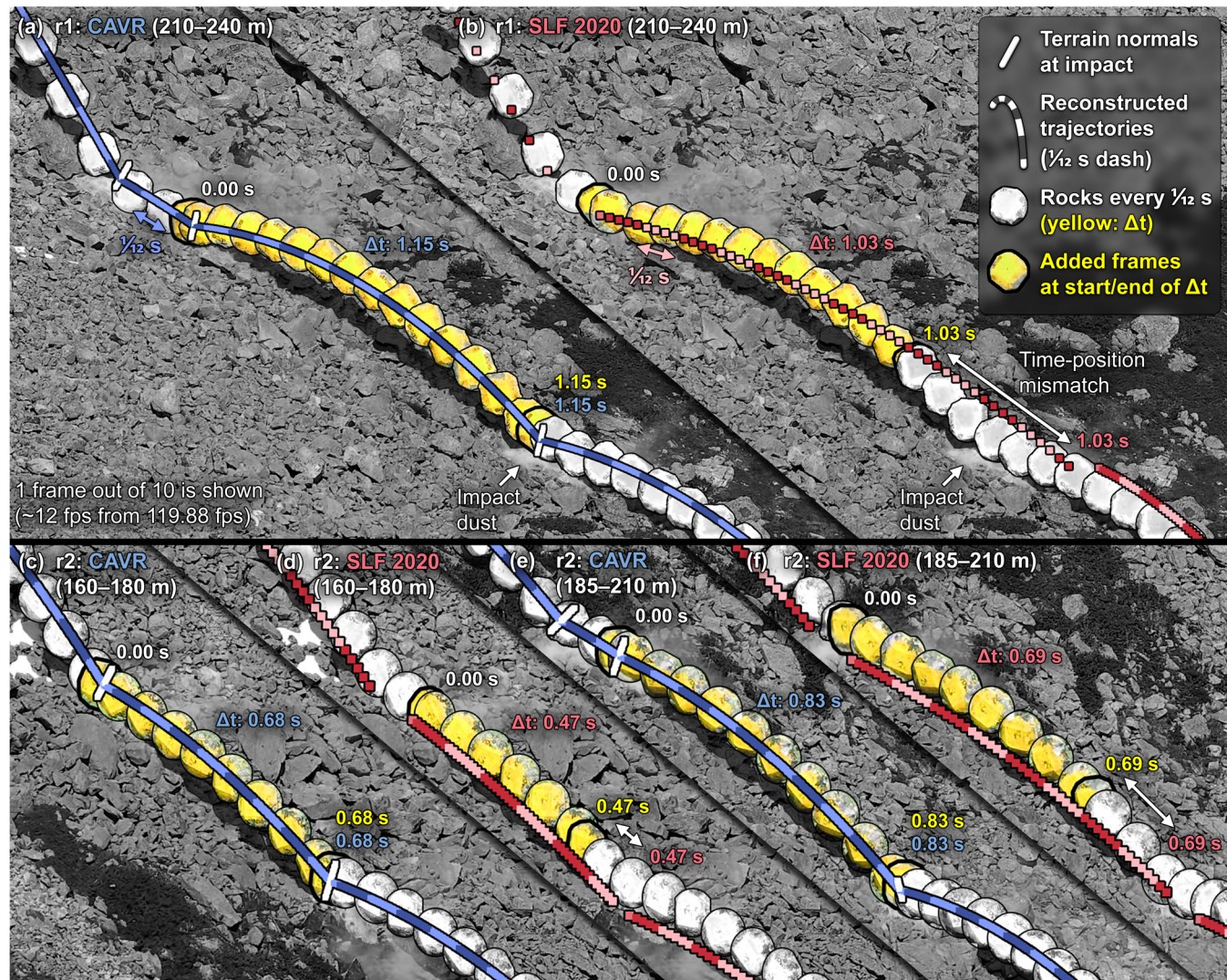

Figure 11: Detailed portion of the three abnormally mismatching reconstructed trajectory segments previously highlighted in Fig. 10e and Fig. 10f. The reconstructed trajectories repeatedly change colors every 1/12th of a second. They are overlaid on the stacked video frames showing the rock's position every 10 frames (every ~1/12th of a second for the footage captured at 119.88 fps). The rocks from the frames corresponding to the chosen period for the three freefalling phases with the energy mismatches are colored in yellow. The visible rocks in yellow from the added frames corresponding to the beginning and the end of the chosen period for the reconstructed parabola are highlighted with a thicker black contour.

For the intercepting impacts at the 2000 kJ flexible ROCCO barrier from Geobrugg, the reconstructed trajectories overlaid with the CAVR method on the stacked video frame using the same viewpoint in Fig. 12 show a good match with the artificial reinforced rocks. The white normal vectors to the terrain automatically calculated are used to obtain a proper offset at the center of mass of the rocks from the impact point picked on the detailed 3D terrain model. With a proper offset and precise timing from 119.88 fps, which is 10 times more frames than those shown in the figures (Fig. 7, Fig. 10, Fig. 11 and Fig. 12), the reconstructed parabolas of the trajectories have heights matching the positions of the real rocks. Thus, the

reconstructed velocities are close to reality. Slight visual misalignments are present in rare occasions, within a margin of approximately ½ of a radius of the related rocks in that case. For example, the 840 kg equant rock at the 5th contact point in-line with the posts from the left appears slightly too much on the left, or the impact with the ground preceding the impact with the fence of the outermost right trajectory is slightly too high. Therefore, these impacts could be refined further, especially if improved by using video footage from other view angles.

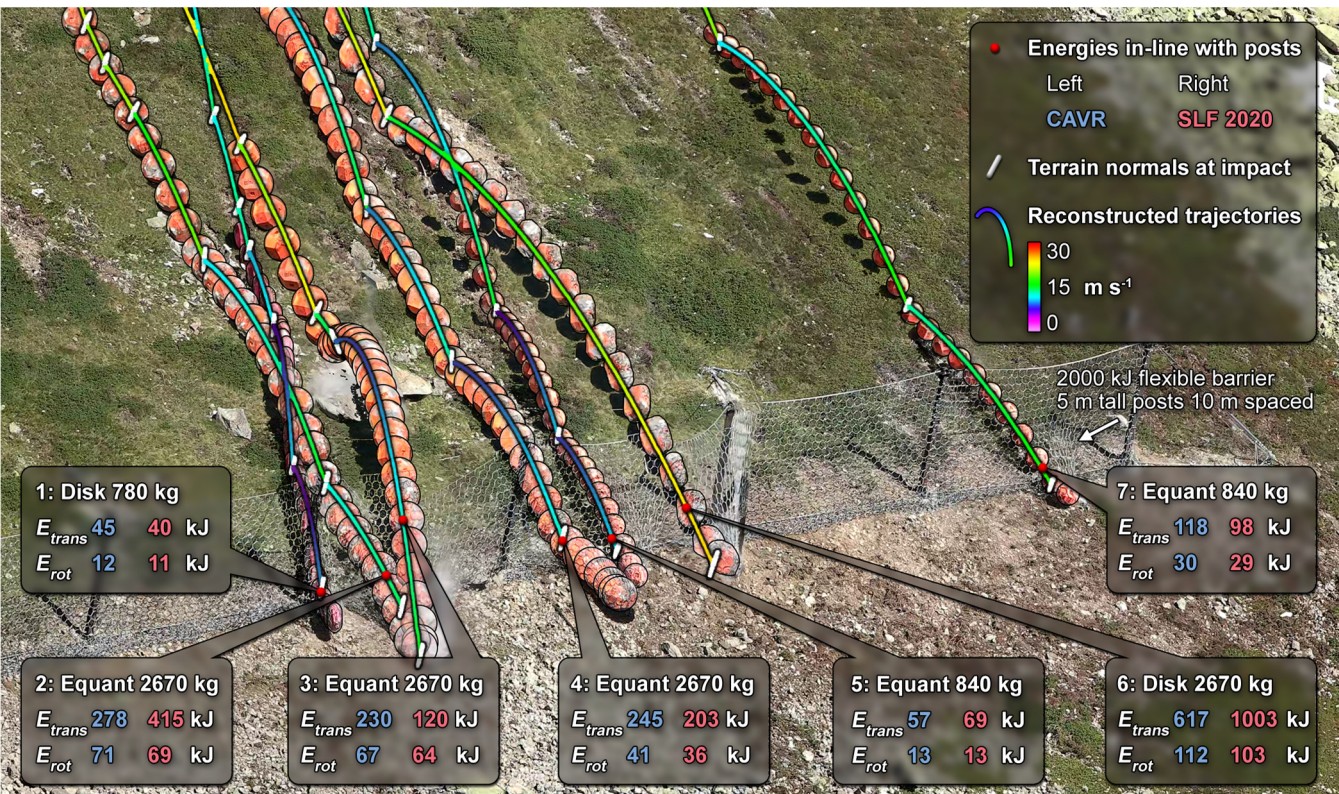

**Figure 12: Reconstructed trajectories using the CAVR method overlaid on the stacked video frames of the different rockfall runs from the SLF experiment performed on 2019 September 13th. One frame out of ten is shown for each trajectory from the 119.88 fps footage captured with the FHD camera setup shown in Fig. 6. All these rocks that are intercepted by the flexible ROCCO barrier from Geobrugg are stopped (Sanchez and Caviezel, 2020). The reconstructed energies in line with the posts from both methods are put side-by-side.**

Despite the slight visual misalignments, the impact positions and heights in line with the posts match the impact fields and points from the 9 sectors in Sanchez and Caviezel (2020). The red points numbered from 1 to 7 from left to right in Fig. 12 correspond to rockfall runs 1.3, 1.9, 1.4, 1.5, 1.2, 1.8 and 1.1, respectively, in Sanchez and Caviezel (2020). The rotational energies with the CAVR method are equal to or slightly above the older values from the SLF. The differences between the two methods are relatively low, with an average of ± 7% and an S.D. of 5% for the seven reconstructed impact values. Thus, the estimated angular velocities from the counted number of rotations during the freefalling periods are equivalent to the SLF values from gyroscopes. Therefore, with the CAVR cost-effective method, it is possible to reconstruct valuable information

even if rocks are not instrumented. The use of gyroscopes can, however, save time during the posttreatment and can be of great help when the rotations of the rocks are not aligned around their principal axes of inertia, highlighting the complementary value of combining different reconstruction methods. The slight differences in rotational energies could be attributed to the use of slightly different moments of inertia. The rock shapes are considered with the CAVR method, but the inhomogeneous mass distribution caused by the denser steel reinforcement of the artificial concrete rocks is ignored. The moments of inertia used for the equant rocks are 62 and 440 kg m$^2$ and 90 and 689 kg m$^2$ for the disk or wheel-shaped rocks. The differences are greater with the translational energy from the two trajectory reconstruction methods. For the smaller rocks approximately 800 kg, they are ± 17% on average with a maximal difference of 21% for the 7$^{th}$ impact from the left in Fig. 12 (1$^{st}$ rockfall run of the day, on the outermost right in the figure). This increases for the four values of the larger 2670 kg rocks with an average difference of ± 46% and a maximal difference of 92% for the 3$^{rd}$ impact from the left (4$^{th}$ rockfall run of the day). These translational energy mismatches are similar to the rare abnormal ones previously covered for the r1 and r2 reconstructed trajectories of the 6$^{th}$ and 7$^{th}$ rockfall runs. They could be attributed to similar timing-positioning imprecisions of the older reconstructed trajectories from the SLF.

With the publicly shared data, it is possible to use the CAVR method for the comparison of the nonoptimal single viewpoint configuration, highlighting the flexibility of the method to handle the variable available footage. It is demonstrated that the reconstructed trajectories align relatively well with the real rock timed positions from the stacked video frames. The velocities and energies also compare well with those from the older reconstructed trajectories of the SLF from the side-by-side comparison. The bounce heights, rotational energies and impact positions against the flexible barrier also compare well with those from the SLF. Therefore, the comparison shows that the presented reconstruction method can reproduce proper 3D rockfall trajectories from experiments or real events, despite some discrepancies observed with the older reconstructed translational energies from the SLF. All methods can be improved, and the CAVR method is no exception. Therefore, opening access to the valuable input data as previously done by the SLF allows the independent review of the data, the combination of different approaches, and the development of innovative solutions. The contribution enables transparent and open rockfall science to be very helpful for the geohazard community when assessing the sensitivity of current rockfall simulation software and for finding the right simulation parameters to be used for similar sites. Hopefully, it will also facilitate the development of more objective rockfall simulation models that are less dependent on inconvenient and expensive back analyses.

**6 Conclusions**

As shown, the implications of the CAVR reconstruction method can be numerous. The reconstructed trajectories and associated information provided can serve three main purposes: 1) The first purpose is facilitating the calibration of rockfall simulations from back analysis, 2) The second purpose is allowing a better understanding of the rockfall and impact dynamics, and 3) the last purpose is helping in the development of new simulation rebound models. The presented flexible

and cost-efficient reconstruction method offers many benefits over automatic tracking methods or frame-by-frame photogrammetry of video footage, especially for reconstructing part of the trajectories of past rockfall events where video footage is not optimal. Indeed, it works with nonoptimal video footage, such as the following:

- blurry footage,
- unstable footage from a handheld camera,
- low resolution footage,
- loss of sight of the rock for some frames,
- low contrast of the falling rock with the background,
- acquired from only one point of view.

Furthermore, the relatively light file handling helps saving time and resources. Indeed, most current computer hardware can easily handle FHD footage. Scrolling through video timelines is not interrupted by frame drops, even at a high bitrate, and does not require a powerful graphics processing unit (GPU). Common affordable camera equipment can capture footage at an FHD resolution and should be combined with a bright telephoto lens with good resolving power and acutance at that resolution for the aperture range that is used. The provided abacus can be used to help plan video and photo acquisitions for similar experiments that rely on remote imagery. It can also be used in other situations to ensure that the inputs for photographic monitoring, photogrammetric models or gigapixel panoramic images have the desired level of detail.

Moreover, the CAVR method can work with large rockfall volumes and high energy values unlikely to be experimented with artificially. The exposure to hazardous slopes is reduced since there is no need to measure the impact positions with GNSS. It does not require time-consuming installation of rock inboard sensors, and thus, is not sensitive to high angular velocity changes or acceleration at impact that could saturate the sensors. Additionally, it is not affected by sensor drift due to the accumulation of measurement errors. As a drawback of not using inboard sensors, it does not provide the fine details, such as the accelerations and changes in angular velocity that occur during the short contacts with the ground. The single point impact information is rather generalized to the form of impulses but with detailed evolution of the freefalling phases, which provides data that fulfills the first purpose of facilitating the calibration of rockfall simulations from the back analysis and the two other main purposes to a certain extent. Instrumented rocks provide complementary valuable information depending on the needs. Therefore, methods should be combined based on the desired advantages when needed.

The computer-assisted trajectory reconstruction with live visual validation of the output parabola and $COR_v$ during the whole reconstruction process ensures that no impact is reconstructed with more energy after the contact than before. This is also helped by the trajectory that is reconstructed close to the center of mass of the block. Thus, the reconstructed parabolas are closer to their true lengths for as long as the impacts are short enough to be simplified as single points. Additionally, the computer-assisted reconstruction simplifies and homogenizes the application of the method to ensure that the work can be spread across a wider range of users.

Concerning the understanding of rockfalls and impact dynamics, as well as helping in the development of new simulation rebound models, the computer-assisted method reconstructs trajectories using an impact detection algorithm that ensures that the geometrical impact configuration is properly measured. The way the terrain is perceived by the rocks relative to their

sizes is measured in the same way as how a rebound model can be applied for simulations. In that optic, further developments will consist of using this reconstruction method to acquire data from a past witnessed large rockfall event and from a collaborative rockfall experiment, which will be analyzed in detail and combined with the reconstructed data from this paper.

**Appendix A: Significance of air drag**

For simplicity, the CAVR method, its ballistic equations and comparison results given as examples in this paper neglect the resistance due to air drag. The method could however estimate the drag force ($F_D$) acting on the rock during freefall using Rayleigh's law as done for rockfall simulations in Noël et al. (2021) with Eq. (A1) as follows:

$$\overrightarrow{F_D} = -\frac{1}{2}\rho_{air}\|\vec{v}\|^2 C_D A \hat{v} ,$$

(A1)

where $\rho_{air}$ is the air density (~1.2 [kg m$^3$]), $C_D$ is the drag coefficient of the rock (~0.9), $A$ is the reference surface of the rock (~ellipse: $\frac{1}{4}\pi d_1 d_3$ [m$^2$]) and $\hat{v}$ is the unit vector in the direction of the rock velocity. The acceleration components from Eq. (4) then become:

$$\overrightarrow{g_t} = \begin{bmatrix} g_{xt} \\ g_{yt} \\ g_{zt} \end{bmatrix} \cong \begin{bmatrix} F_{xD}/m \\ F_{yD}/m \\ -9.81 + F_{zD}/m \end{bmatrix},$$

(A2)

Because the acceleration is not constant anymore, new position and velocity equations could be found by integrating Eq. (A2) over time. Using the Newton-Cotes trapezoidal rule with small time step increments (e.g., $\Delta t/1000$), the rock positions and velocities can be approximated numerically from Eq. (1) and (2) (Fortin, 2016). Doing so, approximated values from reconstructed trajectories considering the air drag can be compared to those neglecting the drag to quantify the error introduced by such omission.

To illustrate the differences induced by air drag, an arbitrary unlikely/unrealistic extreme bounce of a hockey puck sized ellipsoid rock over an impact-to-impact distance ($s$) and slope ($\beta_s$) of 1 km and 25° is shown in Fig. A1. Note that for such illustrative extreme example involving velocities near and beyond the speed of sound, the effect of drag should be event higher due to the compressibility drag dominating at transonic speeds. Despite neglecting such additional drag mostly present at transonic speeds, the differences are sufficient for giving a visual perception of the effect of drag on the reconstructed trajectory that can be pictured behaving similarly to those of shuttlecocks during badminton games, but at different scales. At the starting point, impact a), the returned angle ($\theta_{a2}$) of 29.88° is lowered by 15.21° to a resulting value with drag of 14.67°, so giving a difference of -52 % of the value without drag. The returned translational velocity ($v_{a2}$) is strongly increased by 464.59 m s$^{-1}$ (516 %) to compensate for the important losses during freefall due to drag. The resulting first half of the trajectory with drag appears thus straighter and closer to the ground due to the higher velocities. The maximal vertical bounce height ($f$) and its component perpendicular to the impact-to-impact line ($f_p$) are shifted toward the impact b)

but are reached earlier than $\Delta t/2$ by -1.29 s (-26 %). Inversely at impact b), the incident angle ($\theta_{b1}$) is increased by 19.29° (94 %) while the incident translational velocity ($v_{b1}$) is reduced by 74.01 m s$^{-1}$ (-58 %). Interestingly with this extreme example, the velocity of the rock projectile is maximal at the start and decreases toward its terminal velocity when considering drag.

Without drag, the rock rather accelerates from the apex of its parabola and reach its maximal velocity after losing more than 400 m in height at the impact b).

For more realistic and appliable examples evaluating the differences induced by air drag, the longest parabola and a following smaller parabola of the reconstructed r1 rockfall run corresponding to the 6th run performed by the SLF on September 13th, 2019, at the Chant sura test site are used (Fig. A2a). The highest translational velocity from the 9 runs

performed that day is obtained at the end of the r1 longest parabola. Maximal "realistic" differences can thus be expected with this example, while values closer to the average can be expected with the smaller parabola. The r1 run used an EOTA$_{221}$ 2670 kg wheel/disk shaped artificial reinforced concrete rock of diameter $d_1 = d_2$: 1.45 m and $d_3$: 0.72 m that propagated downslope rotating mostly around its $d_3$ axis. To give a range of differences that can be expected with the presented reconstruction method for similar parabolas, a range of rock sizes going from 0.5 cm to 16 m is tested with and without drag

by scaling up and down the r1 rock. Their related masses range from 0.1 g to 3500 t, and they are all simulated from and to the same parabola bounding points located at the center of mass of the initial 2670 kg rock.

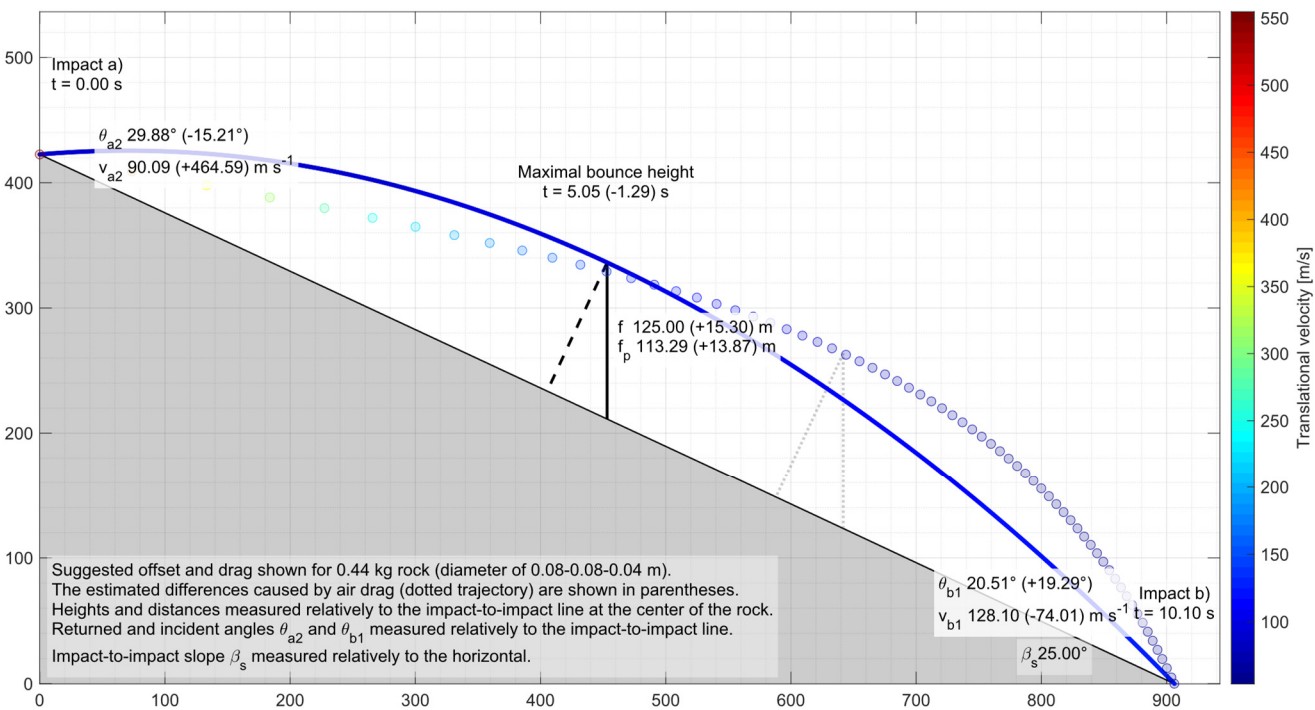

**Figure A1: Unlikely/unrealistic extreme bounce of a hockey puck sized rock over an impact-to-impact distance ($s$) and slope ($\beta_s$) of 1 km and 25°. The dotted trajectory considers the resistance caused by air drag but neglects the compressibility drag. The latter**
**may play an important role in this extreme example however due to the high velocities beyond the speed of sound in the air. The drag is approximated numerically by updating $g_t$ every $\Delta t/1000$ (0.01 s in that case), with one dot of the trajectory shown every 15 iterations.**

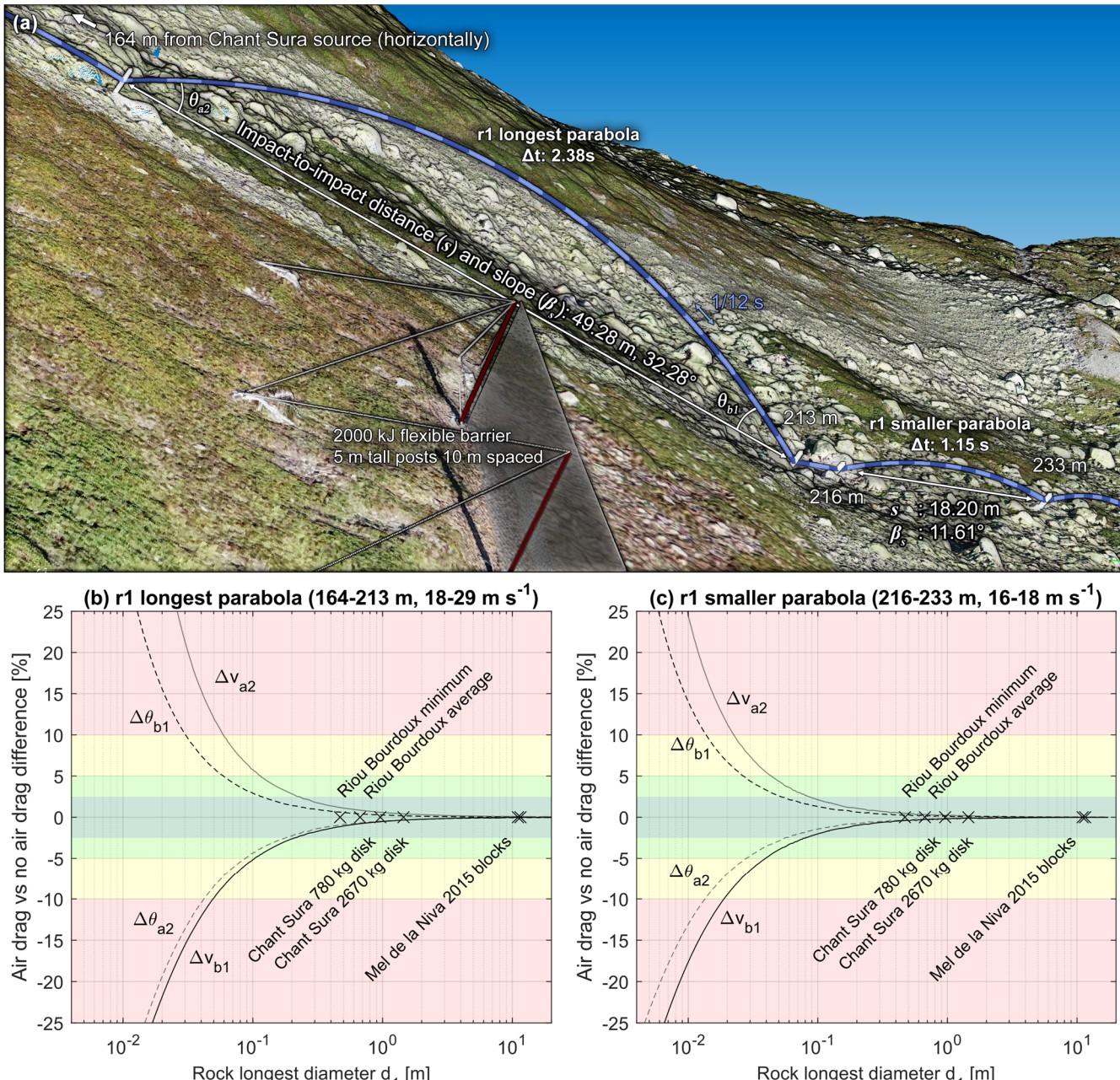

**Figure A2: Chosen parabolas to evaluate the significance of air drag on the reconstructed results and the positioning and timing precision in Appendix B. Note that $\theta_{a2}$ and $\theta_{b1}$ are calculated relatively to the impact-to-impact line in 2D vertical profiles and not in 3D in that case. The differences when comparing the reconstructed incident and returned velocities and angles with vs. without drag for different rock sizes are shown in the graphs. Cross markers are placed as indicative rock $d_1$ diameters as references for the Riou Bourdoux test site (Hibert et al., in review), the Chant Sura test site (Caviezel et al. 2019, 2020, 2021; Sanchez and Caviezel, 2020), and the Mel de la Niva Mountain 2015 rockfall event (Noël et al., in preparation; Lu et al., 2018).**

The ranges of obtained differences when considering drag vs. without drag for reconstructing the trajectory segments of the two chosen parabolas are shown in Fig. A2 and are detailed for the 2670 kg wheel/disk shaped rock of the r1 run in Fig. A3 for its longest parabola and in Fig. A4 for its smaller parabola. Cross markers are placed as indicative rock size references in Fig. A2 with the 780 kg and 2670 kg wheel/disk shaped rocks used at the Chant Sura test site. Markers are also used to indicate the size of the smallest and average rock $d_l$ diameters used at the Riou Bourdoux test site (Hibert et al., in review)

and for the two large rock block fragments that propagated downslope over more than a kilometer in 2015 from the Mel de la Niva Mountain (Noël et al., in preparation; Lu et al., 2018). As for the previous extreme example, the reconstructed initial returned velocities ($v_{a2}$) must be greater to compensate the losses due to air resistance when drag is considered. The trajectories must also be closer to the ground initially, as shown with the lower returned angles ($\theta_{a2}$). The incident velocities ($v_{b1}$) are lowered by drag and come with steeper plunge shown with the higher incident angles ($\theta_{b1}$) as seen previously.

Those differences are however marginal for large rocks like the 2670 kg rock of the r1 run and they are visually imperceptible at Fig. A3 and Fig. A4. For the two chosen parabolas, the differences increase as the size of the rocks decrease. The same would happen with increasing impact-to-impact distances and/or involved velocities.

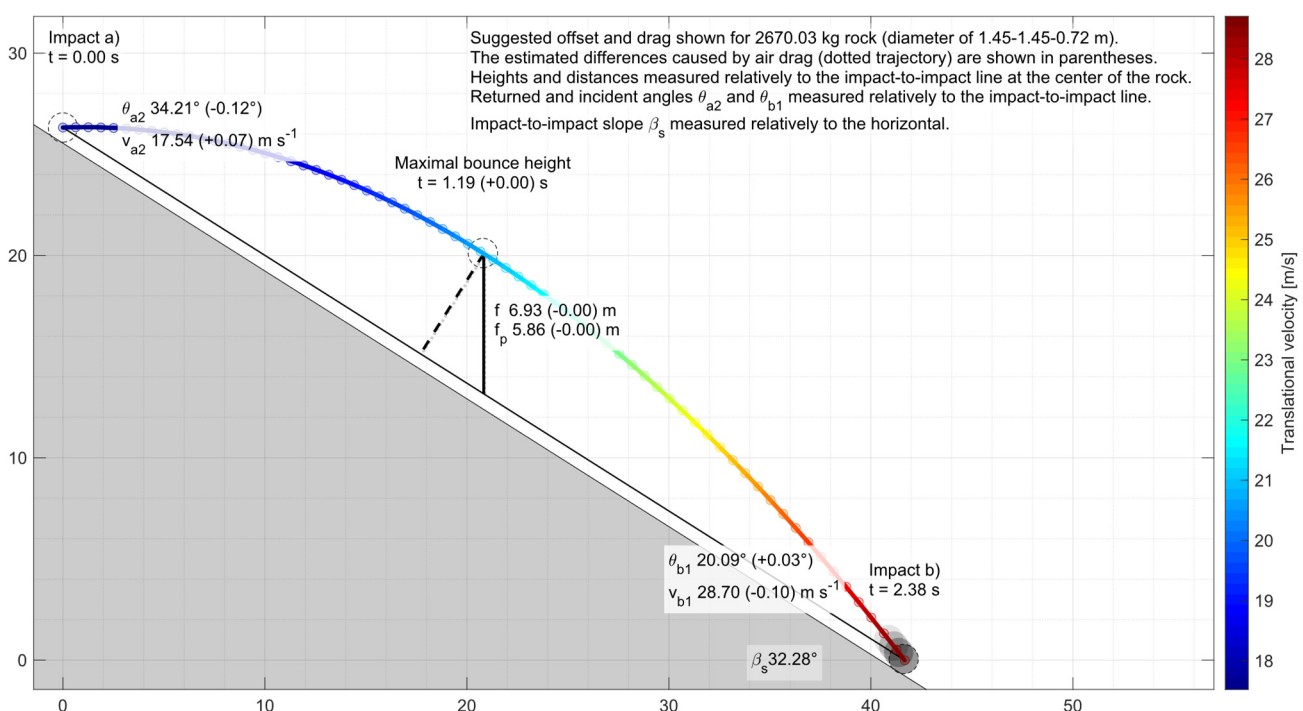

**Figure A3: 2D vertical profile of the longest parabola of r1 reconstructed with and without air drag.**

The obtained differences are lower than ±2.5 % for rocks larger than 0.15 m (>3 kg) in the case of the r1 longest parabola (Fig. A2b), and for rocks larger than 0.09 m (>0.6 kg) in the case of the r1 smaller parabola (Fig. A2c). At such differences, an impact bounded by two similar parabolas would have an apparent coefficient of restitution ($COR_v$) underestimated by 4.9

% when neglecting the air resistance due to the combined error of the incident and returned reconstructed velocities. For impacts bounded by two parabolas like the r1 longest one, the *COR*$_v$ neglecting drag would be respectively underestimated from the smallest to the largest rock markers by 2.3, 1.6, 1.1, 0.8, 0.2 and 0.2 %. For the smaller r1 parabola, the *COR*$_v$ neglecting drag would be respectively underestimated by 0.9, 0.6, 0.5, 0.3, <0.1 and <0.1 %.

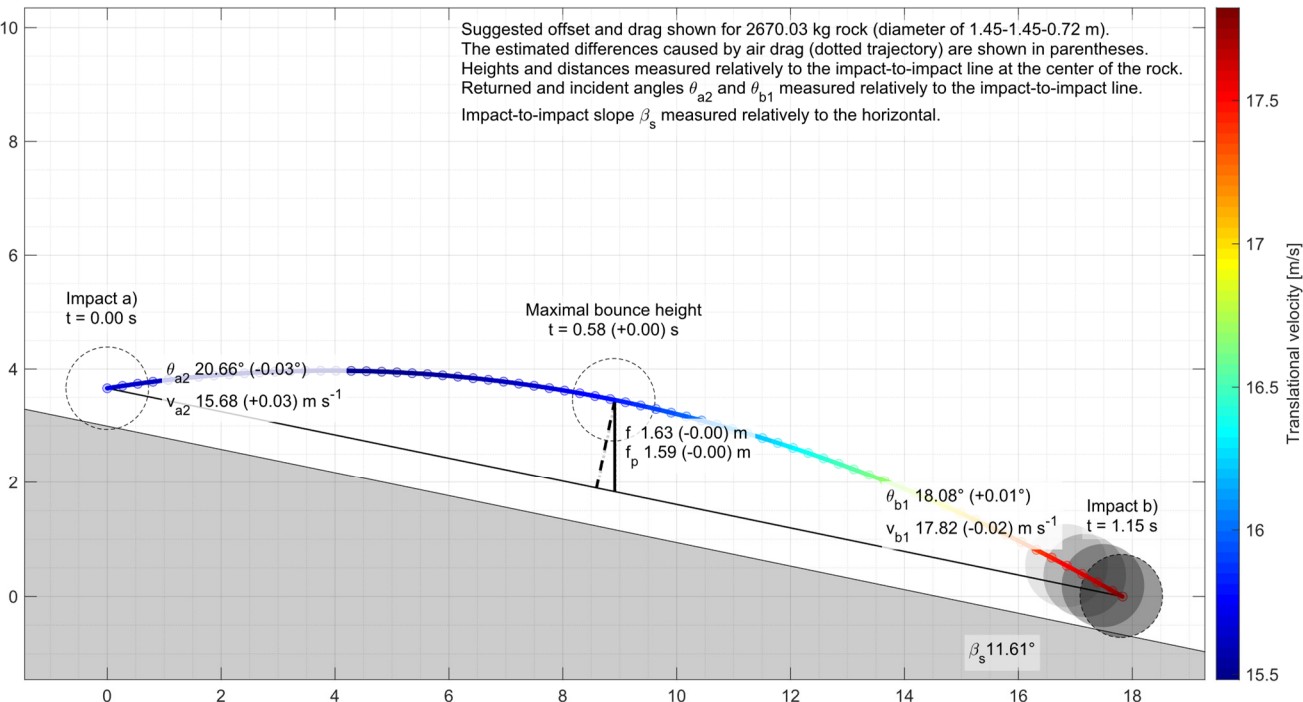

**Figure A4: 2D vertical profile of the smaller parabola of r1 reconstructed with and without air drag.**

As shown, neglecting the effect of the air resistance has most of the time little influence on the obtained results. Still, the effect of air drag might be significant in the case of reconstructing the trajectories of small rock fragments freefalling over long distances at high velocities. In that case, the method can be bonified by implementing the air drag from Eq. (A1) and Eq. (A2) for example.

**Appendix B: Positioning and timing precision**

When the impact position cannot be resolved from visible impact marks on the detailed terrain model and when the impact timing cannot be resolved from rock inboard sensors or proximal geophones, both must be determined visually from the video footage. Sharp and detailed video footage is of great help in that case. When multiple cameras are used, they can be synchronized visually from fast changing objects, like the face of the rotating rock quickly passing from being exposed to sunlight to shadow, or from the quick projection of small fragments. Whatever timing method is used, the error on the

estimation of the freefalling period ($\mathit{\Delta}t$) can significantly affect the reconstructed values. To quantify the effect of the positioning and timing precision, the previously used longest parabola and smaller parabola (Fig. A2a) of the reconstructed r1 rockfall run are here used with different $\mathit{\Delta}t$ and impact-to-impact distances ($s$). The reference impact-to-impact distance and period are respectively set to 49.2830 m and 2.3774 s for the longest parabola and to 18.1957 m and 1.1512 s for the smaller parabola.

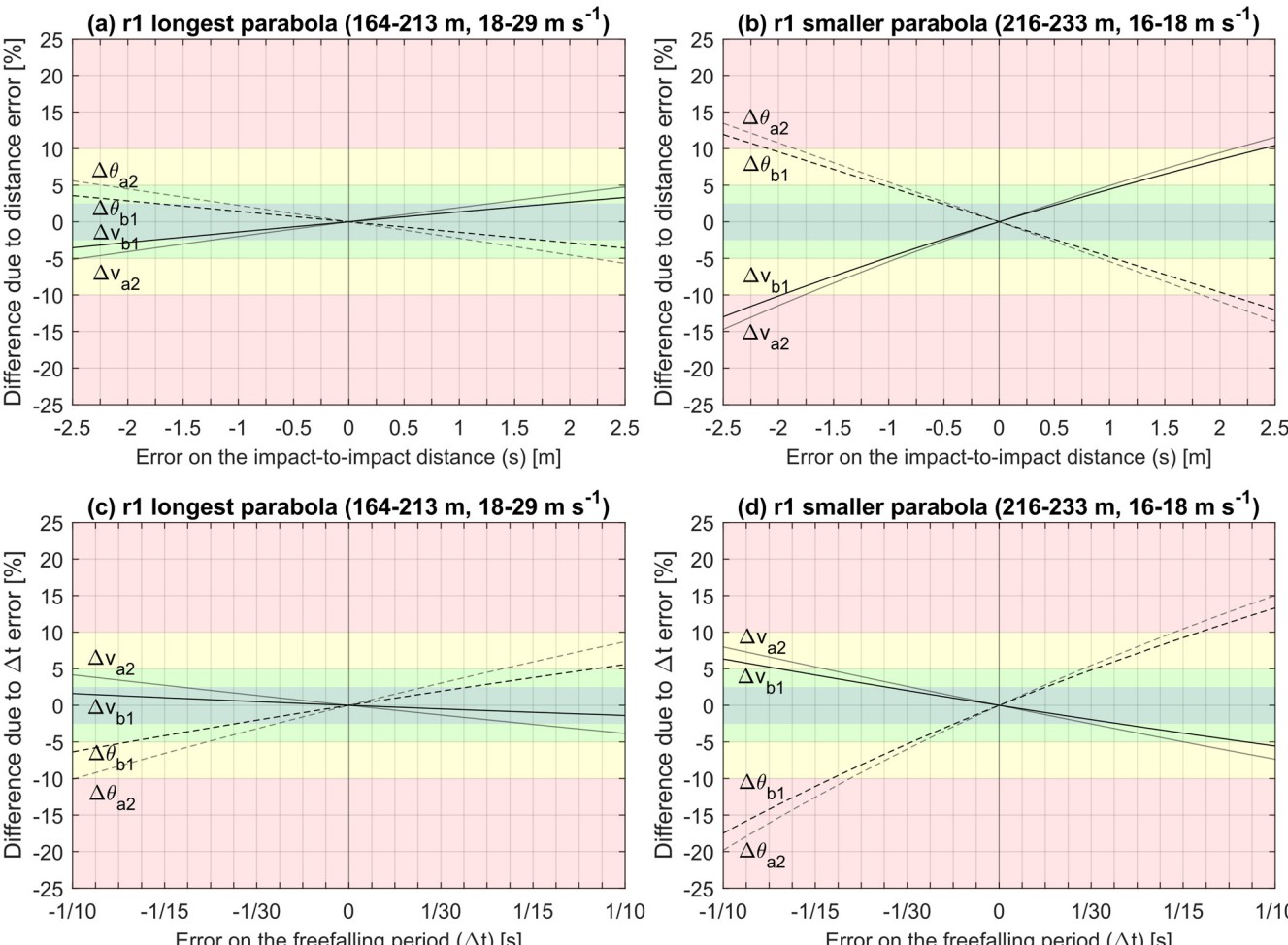

**Figure B1: Differences induced by erroneous positioning or timing shifts for the two reference parabolas of the r1 rockfall run.**

The obtained differences when comparing the reconstructed values at different impact-to-impact distances relatively to their references are shown in Fig. B1a and Fig. B1b. Those related to different $\mathit{\Delta}t$ relatively to their references are shown in Fig. B1c and Fig. B1d. Their vertical grid lines are horizontally spaced by 1/120th of a second to correspond to the frame rate of the camera used given as examples in this paper (119.88 fps). The smaller parabola is more sensitive to the positioning and

timing precision due to its shorter impact-to-impact distance and period. Also, when an impact point is erroneously shifted toward its previous impact point or if its impact time is delayed, the effect is double:

1) Its incident parabola becomes more vaulted and with lower velocities, as shown by the rise of the incident and returned angles and decrease of corresponding velocities due to the shortened impact-to-impact distance or prolongated period.

2) Its returned parabola behaves the opposite due to its extended impact-to-impact distance or shortened period.

In such circumstances, a shift of 0.5 m of an impact bounded by two similar parabolas would overestimate its $COR_v$ by 1.7 %

and 5.0 % respectively for bounding parabolas like the r1 longest parabola and the r1 smaller parabola. An impact time delayed by 1/30 s, which corresponds to four frames at 120 fps or two frames at 60 fps, would overestimate its $COR_v$ by 1.9 % and 4.6 % respectively for bounding parabolas like the r1 longest parabola and the r1 smaller parabola. This highlights the importance of prioritizing sharp, detailed, and high frame rate video footage if the acquisition setup can be customized.

Fortunately, an erroneous position or time shift induces opposite changes in the vaults of the incident and returned parabolas

which can be noticed if pronounced enough during the visual validation and fine-tuning of the results. The impact can then be fine-tuned to balance the bounding parabolas until they match with the observations. This highlights the advantage of the computer-assisted reconstruction where the reconstructed parabolas updated in real-time can be quickly validated without time-consuming intermediate steps.

**Data availability statement**

The video footage, 3D detailed high-resolution terrain models and older reconstructed trajectories are available via Caviezel et al. (2020).

The 3D reconstructed trajectories for the side-by-side comparison are available as supplemental material. As previously mentioned, they can be refined further.

The reconstruction tool can be customized for different rockfall test sites and camera setups and can be freely obtained upon

request to the first author.

The impact-detection algorithm applicable to 3D rockfall simulations from Noël et al. (2021) can be freely obtained via https://stnparabel.org or upon request to the first author.

**Author contribution**

F.N., M.J., A.C., C.H., F.B. and J.P.M. conceptualized the research. F.N. and A.C. oversaw the data curation. F.N. and M.J.

did the formal analysis. F.N. and A.C. contributed to the investigation. F.N., M.J. and A.C. developed the methodology. F.N., M.J. and A.C. oversaw the project administration. F.N., M.J. and A.C. provided the resources. F.N. developed the software. M.J. and A.C. supervised the experiment and the research. F.N., M.J., A.C., C.H., F.B and J.P.M. validated the

reconstruction approach. F.N. designed and produced the figures. F.N. wrote the original draft. F.N., M.J., A.C., C.H., F.B. and J.P.M. reviewed & edited the original draft.

## Acknowledgments

The authors acknowledge A.J.E. for editing the English of the manuscript. A special thanks goes to the SLF and Geobrugg for the collaboration and access to the Chant Sura test site during the rockfall experiment on September 13th, 2019. We thank Synnøve Flugekvam Nordang for her help with the reconstruction of the trajectories with the presented CAVR method and with the design of the figures. Finally, the authors would like to acknowledge the reviewers of the present manuscript.

## Competing interests

The authors declare that they have no conflict of interest.

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
