# Peer review of "Rockfall trajectory reconstruction: A flexible method utilizing video footage and high-resolution terrain models"

_Earth Surface Dynamics, 2022_

## Author Comment (AC1)

**Author's response to the comments received for the manuscript "esurf-2022-16"**

Hereunder is a point-by-point reply to the Referees that reviewed our first submission (esurf-2022-16). The Referee's comments are formatted in *black italic*.

We are assured that the critical and constructive comments provided by the referees lifted the submitted work. We are thankful toward the referees for the time they invested reviewing the manuscript.

Sincerely,

François Noël, on behalf of the authors

Anonymous Referee #1

*The authors present an cost-effective method to reproduce 3D rockfall trajectories from and CAVR. It has the advantage that uses the 3D terrain model as the spatial reference for the falling block instead of the videocameras and considers a proper offset of the center of mass in the reconstruction of the trajectories. Nice sutdy, clear presentation and meaningful results.*

Thank you very much for the time spent reviewing our manuscript. We are glad that the work is positively appreciated. Thank you also for the constructive feedbacks, copied hereunder with our answers. They were addressed in the revised manuscript as described in the answers.

Specific comments

*1) In my opinion, the sentence "only dissipative impacts are observed with CAVR..." should be removed from the summary. The authors to the SLF experiments. The sentence may be misleading because although it refers to the SLF experiment, the reader may interpret it to be generalizable to other experiments.*

Indeed, without specification of which experiments had apparent creation of energy and about what could be the possible causes, we realize that the sentence in the abstract may be misleading thanks to your comment. The sentence has thus been removed.

Reference line (tracked change): 25 (25)

*2) Lines 34-35 For ease of reading, it is convenient to explain here the difference between the rebound model parameters and the apparent coefficient of restitution.*

This is a good point because if misunderstood by the reader, this could lead to a misuse of rockfall simulation models when directly using reconstructed values from rockfall experiments. To clarify the difference in between the apparent coefficient of restitution and the rebound model parameters, we added an example in the first lines of introduction. We chose this approach to also illustrate why both cannot be directly interchanged:

"…Indeed, even if rebound model parameters are classically called "coefficients of restitution" (e.g., $R_N$ and $R_T$ for the normal and tangential components), they are not the same as the apparent coefficients of restitution (e.g., $COR_N$ and $COR_T$ for the respective components) and cannot be directly interchanged, as explained in Noël et al. (2021). For example, a rebound calculated with the model of Pfeiffer and Bowen (1989) using a normal "coefficient of restitution" of $R_N = 0.35$ as damping parameter returns a normal translational velocity of 1.6 m s$^{-1}$ for a normal incident velocity of 10 m s$^{-1}$. In that case, the ratio of returned velocity over the incident one gives a calculated apparent coefficient of restitution of $COR_N = 0.16$, which is different than the $R_N$ of 0.35 used. Because it is difficult to transpose the simulation parameters from experimental results, finding the proper parameter values are thus far more limited to a range of rock sizes, shapes, terrain materials and saturation, perceived roughness, and profile geometry…"

Reference lines (tracked change): 35-43 (36-44)

*3) Section 3.3. As the authors mention, it is necessary to evaluate the geometry of the rock block to adequately compensate the impact positions to the center of mass to obtain the correct trajectories. However, the reconstruction the 3D model of the blocks is not fully described. It seems that the model is obtained from the frames but there may be hidden parts of the blocks that can affect the reconstruction of the shape and volume, as well as the measured dynamic parameters. It is convenient to explain in more detail the procedure followed to reconstruct the volume of the blocks and also provide an estimation of the errors.*

For terrain with changing slope and/or orientation in between two impacts, omitting to offset the impact positions to the center of mass of the rock would change the impact-to-impact distance. We added the Appendix B to the revised manuscript to detail quantitatively the error induced by such change. Additionally, the Appendix A was added to detail

quantitatively the error induced by neglecting the air resistance. The presented method is flexible to different ways of estimating the rock geometric characteristics that come with their respective precision, like from SfM photogrammetry or terrestrial laser survey. To respect the "cost-effective" theme of the method, we added details about quick estimations that can be done effectively from field measurements and simplified rock shapes:

"…The rock geometry can be evaluated from on-field measurements or with 3D models acquired by SfM or by mobile and TLS methods. The mass can be determined from the volume ($V$) of the rock and the volumetric mass density ($\rho_{rock}$) of rock samples (assuming a homogeneous distribution of the mass). First, an estimation of the volume can be done by simplifying their shapes to ellipsoids from measuring the $d_1$, $d_2$, and $d_3$ diameters of the rocks on field, with $d_1$, $d_2$ and $d_3$ being the lengths of the longest, intermediate, and shortest sides, respectively. In that case, the geometric properties are given with the Eq. (22) to Eq. (26) as follows:

$$V = \frac{\pi}{6} d_1 d_2 d_3 , \tag{22}$$

$$m = \rho_{rock} V , \tag{23}$$

$$I_{d3} = \frac{1}{20} m \left( d_1{}^2 + d_2{}^2 \right) , \tag{24}$$

$$I_{d2} = \frac{1}{20} m \left( d_1{}^2 + d_3{}^2 \right) , \tag{25}$$

$$I_{d1} = \frac{1}{20} m \left( d_2{}^2 + d_3{}^2 \right) , \tag{26}$$

where $I_{d1}$, $I_{d2}$, and $I_{d3}$ are the principal moment of inertia when the rock is rotating around the $d_1$, $d_2$, and $d_3$ axes respectively. Such estimation from ellipsoids can be performed efficiently from 3D models containing numerous rocks using the automatic method described in Steer et al. (in review). The precise moments and principal axes of inertia can also be identified from an analysis on the 3D meshed model of individualized rocks (e.g., with MeshLab from Cignoni et al. (2008))…"

Reference lines (tracked change): 391-406 (397-414)

*4) There is an issuet that has not been treated by the authors but that appears in other experiments. It is the presence of dust during the impact. Has dust been generated? If so, how have the authors resolved this circumstance? Is it the algorithm that determines the point of impact and the kinematic parameters?*

This can indeed be a limitation of the method that requires visual identification of the impact positions. For example, in the case of the 2015 rockfall event from the Mel de la Niva Mountain in Switzerland, only the part of the trajectories outside of the initial dust cloud could be reconstructed using the presented method. In the case of the rockfall experiment performed at the Riou-Bourdoux test site in Hibert et al. (in review), each run had to be performed separated by about 10-15 minutes to wait for the dust to settle. However, small dust clouds are helpful to notice an impact with the ground, the video footage can then be rewind to when the dust appear for proper timing. The computer assisting tool helps offsetting the impact point to the center of mass of the rock based on where the user manually transposes the impact on the ground of the 3D detailed terrain model. The tool then reconstructs the parabolas in real time with their translational kinematic parameters based on Volkwein et al. (2011), Wyllie (2014), Glover (2015), and Gerber (2019).

*5) Please, check the references, some references are incomplete or the source can not be easily identified (e.g Berger, 2011; Domaas, 1995; Garcia, 2019; Girardeau-Mountaut, 2006; Sanchez 2020; …)*

Sure. We apologies for the missing information or wrongly formatted references and in-text citations across the manuscript. They were revised following referee's comments and reformatted to reflect the journal's style.

Reference lines (tracked change): 824-930 (834-943)

Anonymous Referee #2

*Thanks for this nice contribution and the approch to significantly improve video analyses of rockfall trajectories. I have more or less only small comments.*

Thank you very much for the time spent reviewing our manuscript. We are glad that the work is positively appreciated. Thank you also for the constructive feedbacks and suggestions, copied hereunder with our answers. They were addressed in the revised manuscript as described in the answers.

**General comments**

*You state that your analysis procedure has some problems with longer lasting blockground contacts. Would it help to assume that such longer contact periods are not reflecting a single contact but two single short impacts right one after another? Often, these "double impacts" are responsible for the cases where the COR > 1.*

Yes, this can be used with the presented method. We also observed for some impact that they are indeed composed of two small rock-ground interactions. The precision for such short reconstructed parabola is however low because of the small impact-to-impact distance relatively to the local accuracy for picking the impact positions. In such situation, the computer assisted real-time update of the reconstructed values is of great help to ensure keeping the $COR_v$ values under one. This also constrain the selection of the freefalling period and the impact-to-impact distance from the selected impact positions by setting a boundary that should not be crossed. Without this constrain, the precision depends on the accuracy at picking the impact positions and timing them, which we detailed with two examples in the added Appendix B to the revised manuscript.

Reference lines (tracked change): 765-798 (776-809)

*Would it be helpful for your purpose to rely on existing data regarding camera/lense distortions such as e.g.*

*https://argus.web.unc.edu/camera-calibration-database/*

*https://lensfun.github.io/*

Given the complementary nature of the presented method, those recommended references can certainly be useful for reconstruction methods relying mostly on the cameras as references. The software (Agisoft Metashape Professional) used by the SLF to produce the detailed 3D terrain models of the Chant Sura test site (Caviezel et al., 2020) also applies lens correction to ensure producing undistorted terrain models. In the case of the presented method, the terrain model is the spatial reference instead of the cameras. It is distorted equally with the manually tracked rock in the forefront of the captured images, thus marginally affecting the picked positions only by distorting the circular picking accuracy and conic sections. If the rocks are kept in the center of the frame where distortion is often minimal, this effect can be neglected with the presented method (but can still be corrected in post-production with the suggested methods, at the same time as potentially adding numerical image stabilization and time code overlay). We clarified this distinction of the presented method relatively to those using the cameras as references in introduction:

"…Contrary to the tracking methods of Dewez et al. (2010) and Caviezel et al. (2019), the cameras can be zoomed to narrow FOVs and move or panned to track the rocks, since the 3D detailed terrain model acts as the spatial reference instead of the cameras. This produces detailed close-up footage of the rocks and the surrounding terrain features that facilitate the visual identification of the impact points with the ground. Lens distortion is less problematic as it shifts equally the captured rocks with their surrounding terrain that act as reference. It also increases the flexibility of the method, as different video footage can be used as input…"

Reference lines (tracked change): 106-111 (108-113)

*If you have another camera with different viewing angle how much does the (probably) missing synchronization of the different camera influence the analysis?*

Interesting question. We added the Appendix B to the revised manuscript detailing how a shift in timing and positioning affects the reconstructed trajectories. The timing is expressed in 1/120 s divisions to help making parallels with erroneous frame shift due to synchronization problem or related to choosing the right frame when the rebound happens. An erroneous shift in time has an opposite effect on the incident and returned parabolas, one becomes straighter while the other becomes more vaulted. Such changes may become visually perceptible when verifying that the reconstructed trajectories match with the video footage. Again, this fine-tuning process is eased with the computer assisted reconstruction where timing and positioning changes can be seen on the parabolas updated in real-time. The method presented is intended to be a complementary or alternative option to existing methods. If more precise timing can be obtained from alternative methods, like from inboard sensors or proximal geophones in case of instrumented setup, one should choose the most precise option available. As for the camera synchronization, we recommend using one camera as main reference for the time and the others as visual support.

Reference lines (tracked change): 765-798 (776-809)

*Your manuscript almost everywhere uses qualitative expressions only such as "low, high, good, well, higher, lower, longer, slightly, close, ….". I would love it to have also some quantitative values as well. Especially in the discussion section. This would lift the quality of the article to a higher level.*

Good observation, we do agree that, when possible, quantitative terms, expressions, tables or graphs should be given in order to obtain a high-level publication. Qualitative expressions were used sometimes because the method described can be applied to different situations and sites where the precision may vary. Indeed, each impact has a unique precision depending on the terrain angle, orientation, distance from the camera(s) and acquisition setup. An abacus is provided to help quantify the level of detail that one may expect from his/her chosen acquisition setup. The impact position picking accuracy is conceptualized to show how it can be quantified. In the revised manuscript, an applied example of the conceptualized picking accuracies where cone sections from multiple cameras meet was added in the form of a new figure (Fig. 4) reproduced hereunder.

[Figure]

*Figure 1: Examples of spatial accuracies from the projection of ±4 px picking accuracies for the Riou Bourdoux rockfall test site that involved multiple ground based and airborne camera viewpoints (Hibert et al. in review). The maximized local precision and accuracy of the 376 constrained areas of the overlapping ellipses are shown in bright green like in the previous figure. The maximal sizes (worst values) of some maximized local accuracies are written in bright green next to their impacts. Note that each impact accuracy is unique and depends on many variables like the impacted terrain geometric configuration and texture, its distance from the viewpoints and the video acquisition setup that influence locally the ease of visually identifying the impact location.*

Reference lines (tracked change): 266-273 (270-277)

Additionally, in the discussion section, qualitative terms are employed when describing visual comparisons in between reconstructed trajectories and the positions of rocks visible in video footage. To clarify on that aspect, we added the term "visual" in the revised manuscript, like here: "being visually aligned with", "Visual alignment mismatch" and "Slight visual misalignments". As for the comparison in between the reconstructed trajectories (r1 and r2) employing the presented method (CAVR method, in blue) vs. those from Caviezel et al. (2020) (SLF 2020, in red), the translational energies and bounce heights are quantitatively compared for every horizontal distance from the source in Fig. 10 also reproduced hereunder. Its description has been enhanced in the revised version of the manuscript to ease its interpretation.

[Figure]

*Figure 2: Side-by-side comparison of the reconstructed translational velocities, positions, translational energies, and bounce heights of the reconstructed trajectories with the CAVR method compared to the older trajectories from the SLF (Caviezel et al., 2020). The r1 and r2 rockfall runs correspond to the 6th and 7th runs performed by the SLF on September 13th, 2019. The 3D reconstructed trajectories are overlayed on the stacked frames showing the rock positions every 10 frames (every ~1/12th of a second for the footage captured at 119.88 fps with the camera setup shown in Fig. 6). For scale, the dimensions of the 2670 kg orange reinforced disk or wheel shaped concrete*

*rock are 1.45 m (d₁ and d₂) by 0.72 m (d₃). The reconstructed trajectory segments are shown in black with their respective slope profiles in gray as background elements of the 2D vertical profiles shown in (e) and (f). The characteristic sawtooth shape of the energy profiles helps distinguish them from the bounce heights. The values from the CAVR method are shown in blue while those from the SLF are shown in red for a quantitative comparison. The portion of reconstructed trajectory segments shown over stacked video frames in (a), (b), (c), (d) and in Fig. (11) are highlighted in the 2D profiles (e) and (f) with white vertical bands.*

Reference lines (tracked change): 534-545 (543-554)

To complement the quantified comparison of the energies and bounce heights in Fig. 10, the quantified spatial difference in between the compared reconstructed trajectories were added in the revised manuscript for the three detailed trajectory segments with abnormal energy mismatch (difference greater than 100 kJ):

*"…In the 3D space, the r1 parabola reconstructed with the CAVR method in Fig. 11a is 2.21 m away in average from the one from the SLF (Fig. 11b), with a S.D. of 0.89 m. Concerning the two r2 parabolas in Fig. 11c and Fig. 11e, they are respectively separated by 0.77 m, S.D. of 0.10 m and 0.79 m, S.D. of 0.02 m from the SLF parabolas (Fig. 11d and Fig. 11f)…"*

Reference lines (tracked change): 567-570 (577-579)

The Appendix A has been added to the revised manuscript to quantify the effect of the omission of the air resistance with the presented method using three examples of reconstructed parabolas.

Reference lines (tracked change): 682-764 (693-775)

The Appendix B has been added to quantify the deviations on the reconstructed values related to the timing and positioning precision reusing two of the three examples from the Appendix A.

Reference lines (tracked change): 765-798 (776-809)

**Specific comments**

*L33: Maybe, you can add references to these two software tools?*

Sure. The references were added in the revised version of the manuscript.

Reference lines (tracked change): 32 (32)

*L36: Maybe, you can emphasize more clearly that the concept of COR in rockfall trajectory modelling probably is a model only. The rocks themself almost don't jump (just let a rock drop and observe almost no rebound). So, the COR approach is not reflecting the correct physics behind but compensates the natural edges and corners of blocks and underground that force the jumps.*

This is a good point also similarly mentioned by the Anonymous Referee #1. The concept of COR in rockfall trajectory modelling surely is a model, like the simplification of impacts with short rock-ground interaction to single points of the presented method. We added an example in introduction to shows how a normal impact is returned with almost no rebound. The example also highlights the differences in between "apparent coefficient of restitution" and "coefficient of restitution" used as damping parameter and why one should be careful at not directly using one for the other. See the author's response to the second specific comment of Referee #1 regarding the details added in introduction to the revised manuscript about the *COR*.

Reference lines (tracked change): 35-43 (36-44)

*L56: The posterior analysis of traces in the field might deliver good restaurations of trajectories. Of course a lot of work but often necessary for example in case of heavy damages. Suggestion of additional publication regarding physical trajectory parabola analyses*

*Gerber, W. (2019). Naturgefahr Steinschlag–Erfahrungen und Erkenntnisse. Eidg. Forschungsanstalt für Wald, Schnee und Landschaft WSL, Birmensdorf. WSL Berichte, 74, 149. https://www.dora.lib4ri.ch/wsl/islandora/object/wsl%3A19475 (in German only, relevant pages 17-22, 33-54)*

Indeed, trajectories should be back calculated from field observations or remotely using drones, total station, or cameras for examples. The interesting publication with methods for back calculation from impact-to-impact distances and slope has been added to the revised manuscript. Additionally, we reformulated to remind on the value of such collected data:

"… From them, Dorren et al. (2005) and Dorren and Berger (2006) used rangefinders with a tiltmeter and a compass to measure the position of each impact, requiring time-consuming and potentially exposed field work to obtain the valuable field data…"

Reference lines (tracked change): 60-62 (62-64)

*L62ff: The advantage of the blocks of Caviezel et al is their regular shape. This eases the determination of the centre of gravity.*

Sure, when the rock shape is known, it can be fitted on its partially reconstructed 3D model at a given time to locate its center of mass. We added such workaround to the revised version of the manuscript:

"Compared to Dewez et al. (2010), this automation process can introduce an erroneous shift of the reconstructed center of mass toward the cameras if the 3D points of the occluded backside of the rocks not visible by the cameras are missing. However, this can be workaround by fitting 3D models of the controlled rock shapes on their partial photogrammetric reconstruction."

Reference lines (tracked change): 75-78 (77-80)

*Caption Fig. 1: "minimize" --> "minimizes".*

Well found! We apologize for this mistake that escaped us. We corrected it as suggested.

Reference lines (tracked change): 140 (142)

*L134: Till which travel speed would you recommend that the air resistance is still neglectable?*

This is a very good question that deserves to be explored further. We added to the revised manuscript the Appendix A about the significance of air drag that we attempted to quantify trough three examples.

Reference lines (tracked change): 682-764 (693-775)

*Equations 10-11: Maybe, you have to explain the difference of these two CORs in details. Most readers usually knot COR_t and COR_n only.*

The example added in introduction in the revised manuscript may help the reader understand the differences in between "apparent coefficient of restitution" and "coefficient of restitution" as damping parameter for rebound models. See the author's response to the second specific comment of Referee #1 regarding the details added in introduction to the revised manuscript about the *COR*.

Reference lines (tracked change): 35-43 (36-44)

*Figure 3: Can you somehow visualize in the figure that the relation between pixels and nature changes also for the different sections of a recorded image due do lense distortions and the geometry?*

Following the answer given to the second general comment, the presented method is not affected much by lens distortion compared to methods where the cameras are the spatial references. Thus, the representation of the effect of lens distortion is not prioritized. We reworked the conceptualization of the spatial accuracy by adding deviated "light rays" of the picking accuracy near the edge of the frame in the figure reproduced hereunder:

[Figure]

Reference lines (tracked change): 217 (221)

*Caption Figure 3: "helps distinguish" --> "helps to distinguish"*

We reworded as suggested

Reference lines (tracked change): 224 (228)

*Equation (21): The letter "a" has already been used for the acceleration. Maybe, you want to change it?*

This is a good observation, as "$a_t$" for acceleration, "$a$" for the first impact and "$a$" for semimajor axis could be source for confusion. Since the air resistance is neglected across the manuscript (apart from the added Appendix A related to the significance of air drag), the acceleration is simply represented by the letter "$g_t$" in the revised manuscript. The semimajor and semiminor axes are fully written.

*L603: "helps save" --> "helps to save"?*

We reworded into "helps saving".

Reference lines (tracked change): 652 (663)

*Figure 9: The two diagrams are difficult to read. I don't know whether I understood them right, but if so then I would recommend:*

- *Remove "segment above"*
- *Integrate Fig.9a,b,c, etc*
- *Remove CAVR and SLF 2020*

The comment is very relevant as the figure's description was too succinct relative to the amount of information given in the figure. The description was revised to better guide the reader as shown in the Fig. 10 reproduced at the last general comment of Anonymous Referee #2. The first and last recommendations were also amended. The authors are not sure to understand the second recommendation about integrating (a), (b), (c), etc.

Reference lines (tracked change): 534-545 (543-554)

*L488 and others: "Fig." --> "Figure" if reference to a Figure is used within the text. Abbreviate "Fig.". only if used within brackets.*

Thanks for noticing the irregularities with the in-text reference to the figures that we unfortunately missed. The "Fig." and "Figure" were revised across the manuscript to reflect the style specified by the journal. A full "Figure" is used if in the beginning of a sentence, "Fig." is used otherwise.

*L535: remove comma after "Caviezel"*

Thanks for highlighting some irregularity with the in-text references. The in-text references were revised across the manuscript and should now respect the style specified by the journal.

*Suggestion of additional references regarding video analyses in the field of rockfalls but using definitively "antique" techniques:*

*Glover, J., Denk, M., Bourrier, F., Volkwein, A., & Gerber, W. (2012, April). Measuring the kinetic energy dissipation effects of rock fall attenuating systems with video analysis. In 12th Congress INTERPRAEVENT (Vol. 1, pp. 151-160). http://www.interpraevent.at/palmcms/upload_files/Publikationen/Tagungsbeitraege/2012_1_151.pdf*

*Glover, J. M. H. (2015). Rock-shape and its role in rockfall dynamics (Doctoral dissertation, Durham University). http://etheses.dur.ac.uk/10968/*

*Volkwein, A., Brügger, L., Gees, F., Gerber, W., Krummenacher, B., Kummer, P., ... & Sutter, T. (2018). Repetitive rockfall trajectory testing. Geosciences, 8(3), 88. https://www.mdpi.com/2076-3263/8/3/88/htm*

Very pertinent and interesting references. We added them across the revised version of the manuscript.

**References**

*The first two references contain author's institutions in brackets. I think these can be removed?*

The authors institutions were removed in the revised manuscript accordingly.

Reference lines (tracked change): 823-827 (834-838)

*Volkwein et al (2011) is mentioned a couple of times in the text but is missing in the references list. Please, add.*

Sorry for this omission, we added the reference to the revised manuscript.

Reference lines (tracked change): 923-925 (937-939)

*References Garcia (2019) and Sanchez+Caviezel (2020) miss information on where these references can be found. Please, enhance.*

The qualificators "Ph.D. Thesis" and "Grenoble Alpes University" were added to Garcia's reference.

Reference lines (tracked change): 866-867 (877-878)

Sanchez and Caviezel (2020) was also detailed similarly. Thanks for noticing the missing information.

Reference lines (tracked change): 906-907 (918-919)

*The usage of "last accessed" currently is inconsistent with the different links provided.*

The references were revised following referee's comments and reformatted to reflect the journal's style